# Therapeutic Strategies to Overcome Fibrotic Barriers to Nanomedicine in the Pancreatic Tumor Microenvironment

**DOI:** 10.3390/cancers15030724

**Published:** 2023-01-24

**Authors:** Hiroyoshi Y. Tanaka, Takuya Nakazawa, Atsushi Enomoto, Atsushi Masamune, Mitsunobu R. Kano

**Affiliations:** 1Department of Pharmaceutical Biomedicine, Graduate School of Medicine, Dentistry and Pharmaceutical Sciences, Okayama University, 1-1-1 Tsushima-naka, Kita-ku, Okayama-shi 700-8530, Okayama, Japan; 2Department of Pharmaceutical Biomedicine, Graduate School of Interdisciplinary Science and Engineering in Health Systems, Okayama University, 1-1-1 Tsushima-naka, Kita-ku, Okayama-shi 700-8530, Okayama, Japan; 3Department of Pathology, Graduate School of Medicine, Nagoya University, 65 Tsurumai-cho, Showa-ku, Nagoya-shi 466-8550, Aichi, Japan; 4Division of Gastroenterology, Graduate School of Medicine, Tohoku University, 1-1 Seiryo-machi, Aoba-ku, Sendai-shi 980-8574, Miyagi, Japan

**Keywords:** pancreatic cancer, tumor microenvironment, nanomedicine, fibrosis, extracellular matrix, fibroblast

## Abstract

**Simple Summary:**

Pancreatic cancer is difficult to treat. Novel treatment strategies are urgently needed to improve the survival rate, which is approximately 10% five years after diagnosis. The use of nanomedicines, which are formulated within a characteristic size range that favors its specific delivery to the diseased tissue, is being actively explored in cancer treatment. However, fibrosis (the abnormal accumulation of a cell type called fibroblasts and the fibrous protein network that they create) is characteristically seen in pancreatic cancer and hinders the delivery of nanomedicines into cancerous tissue. The decreased efficiency of delivery limits the therapeutic effects of nanomedicine in pancreatic cancer. We call this the “fibrotic barrier” to nanomedicine. To overcome the fibrotic barrier, we could target the fibrotic process and/or optimize the nanomedicine design. In this review, we give a detailed overview of strategies to overcome the fibrotic barriers in pancreatic cancer and highlight key gaps in our understanding.

**Abstract:**

Pancreatic cancer is notorious for its dismal prognosis. The enhanced permeability and retention (EPR) effect theory posits that nanomedicines (therapeutics in the size range of approximately 10–200 nm) selectively accumulate in tumors. Nanomedicine has thus been suggested to be the “magic bullet”—both effective and safe—to treat pancreatic cancer. However, the densely fibrotic tumor microenvironment of pancreatic cancer impedes nanomedicine delivery. The EPR effect is thus insufficient to achieve a significant therapeutic effect. Intratumoral fibrosis is chiefly driven by aberrantly activated fibroblasts and the extracellular matrix (ECM) components secreted. Fibroblast and ECM abnormalities offer various potential targets for therapeutic intervention. In this review, we detail the diverse strategies being tested to overcome the fibrotic barriers to nanomedicine in pancreatic cancer. Strategies that target the fibrotic tissue/process are discussed first, which are followed by strategies to optimize nanomedicine design. We provide an overview of how a deeper understanding, increasingly at single-cell resolution, of fibroblast biology is revealing the complex role of the fibrotic stroma in pancreatic cancer pathogenesis and consider the therapeutic implications. Finally, we discuss critical gaps in our understanding and how we might better formulate strategies to successfully overcome the fibrotic barriers in pancreatic cancer.

## 1. Introduction

Pancreatic cancer accounted for 2.5% of new cancer diagnoses and 4.5% of cancer deaths worldwide in 2018 [1]. There is some inter-regional variation in incidence rates: the highest age-standardized incidence rate is in Europe (7.7 per 100,000 people), which is followed by North America (7.6), and Oceania (6.4) [2]. The incidence of PDAC increases with age and is somewhat more common in men than women. Of concern, incidence and mortality are projected to rise quite steadily [2].

Approximately 90% of pancreatic cancers are pancreatic ductal adenocarcinomas (PDAC) [3,4], so the terms will be used interchangeably in this review. While surgery remains central to treatment, less than 20% of PDAC patients have surgically resectable disease largely due to a lack of specific symptoms, especially in the early stages of the disease [3,5,6]. Chemotherapy thus plays an important role in PDAC treatment. Standard-of-care chemotherapy is a regimen consisting of folinic acid (leucovorin), 5-fluorouracil, irinotecan, and oxaliplatin (FOLFIRINOX) in patients with both metastatic [7] and resected [8] PDAC. Albumin-bound paclitaxel (nab-paclitaxel) plus gemcitabine is also effective in metastatic PDAC, and nab-paclitaxel was the first nanomedicine to be approved in PDAC therapy [9]. FOLFIRINOX is considered the more challenging regimen, and therapy is usually selected by assessing the patient’s ability to adhere to the treatment schedule [6]. A head-to-head comparison between FOLFIRINOX vs. nab-paclitaxel + gemcitabine has not been performed, but recent propensity score analyses suggest that either FOLFIRINOX is more effective and cheaper [10,11] or that both regimens have similar effectiveness [12,13]. Second-line treatment options are also expanding. For example, nanoliposomal irinotecan with 5-fluorouracil and folinic acid extends survival in patients with metastatic PDAC who previously received gemcitabine-based therapy [14]. Nanoliposomal irinotecan was approved by the United States Food and Drug Administration in 2015 and is the second nanomedicine available for use in PDAC treatment. Various targeted approaches, driven by an increased understanding of oncogenic drivers and dependencies in PDAC, are also under investigation [15]. Despite these advances, however, 5-year survival remains at approximately 9–11% [1,2,6]. PDAC thus remains one of the most lethal malignancies.

## 2. Fibrotic Barriers to Nanomedicine in the PDAC Tumor Microenvironment

The *enhanced permeability and retention (EPR) effect* was first reported in 1986 by Matsumura and Maeda [16]. The EPR effect theory posits that the functional immaturity of tumor neovessels allows the selective extravasation of macromolecules (*enhanced permeability*). The extravasated macromolecules then remain in tumor tissue due to the underdevelopment of tumor lymphatics (*enhanced retention*). As such, the EPR effect has served as the theoretical rationale for the development of cancer nanomedicine [17,18]. Although much is still left to learn about the EPR effect [19,20,21,22], it is increasingly clear that the EPR effect may be insufficient, especially in clinical settings, to satisfactorily induce a therapeutic response [23,24,25,26,27]. The shortfall of the EPR effect is often ascribed to heterogeneity: not all tumor vessels are leaky [27]. Leakiness seems to be temporally dynamic, as nanoparticles administered at different times do not co-localize within the tumor tissue [21,28]. The occlusion of blood vessels by thrombi may also cause heterogeneity in the EPR effect [29]. A large body of literature exists on strategies for augmenting the EPR effect to improve nanomedicine extravasation [29,30,31]. However, stromal barriers within the tumor microenvironment (TME) further attrite the efficacy of nanomedicine even after extravasation [32,33].

Analyzing the architecture of the tumor tissue in question is important to understanding the stromal barriers to nanomedicine penetration present. Most cancer nanomedicines are intravenously administered, so considering the path the nanomedicine inside blood vessels must cover to reach tumor cells is especially instructive [32]. A simple but useful classification of tumors focuses on the microenvironment in which tumor blood vessels are situated: in tumors with a *tumor vessel phenotype*, blood vessels exist close to tumor cells (Figure 1A), whereas in tumors with a *stroma vessel phenotype*, blood vessels are distanced from tumor cells by stromal tissue (Figure 1B) [34]. Prominent fibrosis is a histopathological hallmark of PDAC and often occupies 40–80% of the total tumor area [35]. Tumor cells and blood vessels exist embedded within this densely fibrotic tissue. PDAC is thus characteristically of a stroma vessel phenotype. Indeed, analyses of PDAC surgical specimens revealed that the median “thickness” of fibrotic tissue, defined as the distance between blood vessels to tumor cells, is around 10–30 μm regardless of tumor stage [36,37]. The thickness of the fibrotic tissue is the distance an intravenously administered nanomedicine must travel at a minimum to reach tumor cells. Therefore, fibrosis is a barrier to nanomedicine penetration of PDAC tissue, and we refer to this as a *fibrotic barrier* to nanomedicine [38]. Although increasingly clear that fibrosis is orchestrated via an interplay of various cell types, fibrotic tissue consists chiefly of quantitatively and qualitatively abnormal fibroblasts and the extracellular matrix (ECM) that fibroblasts abundantly secrete and deposit [39,40,41,42,43].

### 2.1. Fibroblasts: The Key Cellular Mediator of Fibrosis

Ernst Ziegler first proposed the term *fibroblasts* to describe cells that produce connective tissue during healing in 1895, but the description of these cells as “spindle-shaped cells of the connective tissue” by Rudolf Virchow goes further back to 1858 [44]. In stark contrast to this simple definition, the ontogeny of fibroblasts is quite complex [45]. Moreover, it has only recently gained a wide appreciation that the fibroblast executes a myriad of functions apart from producing ECM to generate and maintain connective tissue (reviewed in [41,44,45]). Reflecting such a multifunctional nature, analyses increasingly at single-cell resolution are revealing great fibroblast heterogeneity [44,45].

Fibroblasts play an important role in cancer, where they acquire phenotypic alterations and are referred to as *cancer-associated fibroblasts* (CAFs) [46,47,48]. Studies initially reported genomic alterations driving the CAF phenotype, but it is now generally accepted that they were artifacts and that CAFs are non-mutant cells [48,49]. While tumor cells may undergo epithelial-to-mesenchymal transition (EMT) and demonstrate myofibroblastic features, expert consensus advises considering this a separate entity from CAFs [48]. Traditionally, the transition from a normal (or *quiescent*) fibroblast to a CAF was determined mainly by the acquisition of a contractile (expression of α-smooth muscle actin (αSMA)), synthetic (synthesis of ECM proteins), and proliferative phenotype [46]. This process is usually referred to as *fibroblast activation*, which is a concept drawn from parallels in wound healing [43,46,48]. This is consistent with the notion of tumors as wounds that do not heal [50]. 

Fibroblast activation involves various signaling pathways such as transforming growth factor-β (TGFβ), receptor tyrosine kinase (RTK) signals (e.g., platelet-derived growth factor (PDGF), fibroblast growth factor (FGF), etc.), G-protein coupled receptor (GPCR) signals (e.g., lysophosphatidic acid, sonic hedgehog (SHH), extracellular proteases, etc.), inflammatory signals (e.g., interleukin (IL) 1, IL6, tumor necrosis factor-α (TNFα), etc.), cellular stress signals (e.g., DNA damage, reactive oxygen species (ROS), disrupted metabolism, etc.), ECM signals, and contact signals (e.g., Notch and Eph-Ephrins) (Figure 2) [43,47,48,51]. PDAC cancer cells also secrete extracellular vesicles (EVs) carrying various cargo, including microRNAs such as miR-155 and miR-1290, which play a role in fibroblast activation into CAFs [52,53,54]. Fibroblasts furthermore become activated in response to microenvironmental cues such as hypoxia, lactic acidosis, and mechanical stimuli [55,56,57,58]. As we further explicate in Section 3.2, this notion of an activated, αSMA-positive and ECM-producing CAF mainly captures CAFs with myofibroblastic features. However, not all CAFs are myofibroblastic. Furthermore, αSMA might be an inconsistent marker of fibroblasts responsible for collagen production in fibrosis [59]. Indeed, CAFs are highly heterogeneous and demonstrate great molecular and functional diversity [47,48,60,61,62,63,64].

The heterogeneity of CAFs partly stems from the diverse origins of CAFs in the fibrotic PDAC TME. Pancreatic stellate cells (PSCs) are widely believed to be the major contributors of CAFs in PDAC [65,66,67,68]. On the contrary, a recent report utilizing lineage-tracing suggests that PSC-derived CAFs are numerically minor, albeit with non-redundant functions in shaping the PDAC TME [69]. Another recent lineage tracing study points to the splanchnic mesenchyme as the tissue-of-origin of CAFs in PDAC [70]. Whatever the relative contributions, various non-PSC cell-of-origins of CAFs have been reported to date. These include mesenchymal stem cells [71], endothelial cells [72], mesothelial cells [73], and macrophages [74]. The CAF phenotype, furthermore, is highly plastic and is dynamically shaped by various intercellular and cell–ECM interactions [47,48,60,61,62,63,64,75]. As a result, CAF subpopulations often cannot be clearly distinguished since a singly specific marker does not exist. Even with a combination of markers, expression profiles generally overlap [43,48,60,61,75]. Therefore, in this review, we use “fibroblasts” (or “CAFs” to emphasize the contrast with normal fibroblasts) as an umbrella term for spindle-shaped cells within the fibrotic PDAC stroma, regardless of cell-of-origin. When describing the literature, however, we respect the terminology adopted in the original article (i.e., if the original study refers to PSCs, we too refer to the cells as PSCs).

### 2.2. The ECM in PDAC

The ECM comprises a “core matrisome” of approximately 300 proteins, in addition to ECM-modifying enzymes, ECM-binding growth factors, and other ECM-associated proteins [76]. The core matrisome consists of collagens, proteoglycans (e.g., heparan sulfate, hyaluronan, and versican), and glycoproteins (e.g., fibronectin, laminin, elastin, and tenascins) which demonstrate characteristic domain architectures [76]. The structure and function of these matrisome proteins undergo extensive modification [76,77]. Not only is ECM content increased in the fibrotic PDAC TME, but matrisome composition is also altered in the fibrotic PDAC TME compared to healthy pancreata. Progressive changes in matrisome composition are seen throughout PDAC tumorigenesis, some of which emerge early on in the pancreatic intraepithelial neoplasm (PanIN) stage [78]. CAFs are generally considered the major ECM producers in PDAC, but other cell types including tumor cells also produce ECM and in certain cases may even be better therapeutic targets [78,79].

The ECM is not a static structure. Various ECM remodeling processes are dynamically orchestrated to maintain tissue homeostasis and integrity [80,81]. For example, collagens are first produced as procollagens with propeptides at both N- and C-termini. Three procollagen chains are then aligned (registered) via the C-terminal propeptides after which the characteristic triple helix is formed with the help of molecular chaperones. The propeptides are proteolytically cleaved during or following the secretion of the triple helix [77,82]. Secreted collagens are subjected to further post-translational modifications such as cross-linking, hydroxylation, citrullination, carbamylation, and glycation [77,83]. Furthermore, collagens are engaged by cellular receptors which transduce cellular forces that can deform individual collagen molecules as well as align ECM fibers [81,83]. Finally, collagens undergo turnover, and intracellular and extracellular pathways of degradation exist [82]. Extracellularly, collagens are degraded by collagenolytic enzymes such as the various matrix metalloproteinases (MMPs) as well as cathepsins. Alternatively, collagens are internalized via phagocytosis, receptor-mediated endocytosis, and/or macropinocytosis after which they are lysosomally degraded [82]. Details will vary between different ECM molecules, but as just illustrated with collagen, ECM remodeling can conceptually be divided into three phases: ECM deposition/post-translational modification, force-mediated ECM modification, and ECM degradation (Figure 3) [83]. All of these processes are characteristically deregulated in cancer [83,84].

Matrisome proteins engage various intracellular signaling networks and regulate diverse biological processes such as cellular function and fate in both health and disease [76,85,86,87,88,89]. Cells possess receptors through which ECM proteins and their bioactive fragments signal [85,90,91]. Arguably the most important class of ECM receptors is the integrins. Integrins act as heterodimers consisting of an α-subunit (18 types) and β-subunit (8 types) of which at least 24 unique combinations are known [92,93]. Distinct subunit combinations confer substrate specificity to the integrin heterodimers. For example, integrins α1β1, α2β1, α10β1, and α11β1 recognize fibrillar collagens; α4β1, α5β1, α8β1, αVβ1, αVβ3, and αVβ6 recognize fibronectin; while α1β1, α2β1, α3β1, α6β1, α6β4, α7β1, and α10β1 recognize laminins [94]. Integrins, through the recognition of diverse ECM proteins and subsequent engagement of various intracellular signaling pathways, play a critical role in practically every step of tumorigenesis and progression [92,93,94,95,96,97]. ECM signaling through various receptors such as integrins shapes CAF phenotype and function, thus forming a signaling loop critical in the progression of fibrosis in the PDAC TME [98]. Abnormalities at practically all levels of the multi-layered regulation of the ECM in the PDAC TME offer various opportunities for therapeutic targeting (detailed in Section 3.4) [99,100].

## 3. Therapeutic Strategies Targeting the Fibrotic Barriers to Nanomedicine

In this section and the next, we discuss therapeutic strategies to target and overcome the fibrotic barriers in PDAC. In theory, one can either therapeutically target and modify the fibrotic PDAC stroma to render it more amenable to effective nanomedicine delivery (strategies #1–#4, discussed in this section) or tune various design parameters of the nanomedicine formulation to achieve optimal delivery and efficacy (strategies #5–#8, discussed in Section 4). In reality, a combination of both to varying extents is likely necessary to achieve clinical benefit (Figure 4). 

To target and modify the fibrotic PDAC stroma, one can quantitatively reduce fibrotic stromal content (strategy #1), target fibroblast abnormalities to qualitatively alter fibroblast phenotype (strategy #2), target metabolic processes in fibroblasts involved in the fibrotic process (strategy #3), and/or target ECM abnormalities prevalent in the PDAC TME (strategy #4). These strategies are not mutually exclusive, and a therapeutic strategy intended to target a particular aspect of the fibrotic barrier may inadvertently affect other aspects [101]—in some cases favorably and in others detrimentally.

### 3.1. Strategy #1: Stromal Ablation—Reducing Fibrotic Stromal Content

The simplest strategy to target fibrotic barriers to nanomedicine in PDAC is to reduce fibrotic stromal content altogether (Figure 5). This is known as *stromal ablation* (or *stromal depletion*). A well-known attempt at stromal ablation targeted SHH signaling. The SHH signaling pathway is activated throughout and plays multifaceted roles in PDAC progression [102]. SHH signaling hyperactivation is driven by activating mutations in *KRAS*, which is observed in the large majority of PDAC patients [103,104]. Tumor cell-secreted SHH activates paracrine signaling in stromal cells, notably fibroblasts, and promotes fibrotic changes such as myofibroblastic differentiation and increased ECM deposition [105,106,107]. SHH signaling inhibition with IPI-926 decreased fibrotic stromal content in a genetically engineered mouse model (GEMM) of PDAC, the *KPC* (*LSL-Kras^G12D/+^; LSL-Trp53^R172H/+^; Pdx1-Cre*) mouse model [108], thereby increasing intratumoral delivery of gemcitabine and achieving a transient stabilization of disease [37]. More recently, the co-delivery of SHH inhibitors (cyclopamine or GDC-0449) with paclitaxel as micelles [109] or with SN38 (the active metabolite of irinotecan) as polymeric nanoparticles [110] was shown to improve survival in mice compared to the respective cytotoxic agents delivered as nano-formulations alone. The above results suggest that stromal ablation indeed improves nanomedicine delivery and, as a consequence, efficacy in experimental models of PDAC. 

Other signaling pathways have also been targeted. TGFβ is considered the “master regulator” of, and thus an important therapeutic target in, fibrosis [111,112]. In an experimental model of breast carcinoma, TGFβ ligand neutralization reduced intratumoral collagen I content and improved the delivery of conventional chemotherapeutics as well as nanomedicine [113]. TGFβ also promotes fibrosis in PDAC [114]. The coordinated activation of SMAD2/3 downstream of TGFβ together with Yes-associated protein (YAP) drives fibroblast activation in response to interactions with PDAC cancer cells [35]. TGFβ signaling through Rho-associated kinase (ROCK) in PSCs drives collagen deposition [36]. The TGFβ receptor inhibitor galunisertib/LY2157299 reduced collagen I content in the fibrotic stroma of PDAC in *KPC* mice [115]. Moreover, treatment with the TGFβ receptor inhibitor LY2109761 together with gemcitabine prolonged survival in orthotopic PDAC (L3.6pl inoculated) mice compared to gemcitabine alone [116]. However, given the cytostatic effect of TGFβ on epithelial cells, the therapeutic benefit of directly targeting TGFβ might depend on the mutational status of TGFβ receptors [117]. 

The direct targeting of TGFβ is generally considered to be clinically challenging and/or infeasible due to its myriad functions across multiple organ systems [111,112]. On the other hand, the diverse molecular mechanisms regulating TGFβ signaling offer additional opportunities for therapeutic targeting [118]. Indeed, various attempts have been made in this regard [119]. The angiotensin receptor blocker losartan negatively regulates TGFβ at various levels of the signaling pathway [120]. Losartan treatment reduced stromal collagen and hyaluronan production in orthotopic PDAC (AK4.4 or L3.6pl inoculated) mice and alleviated the solid stress within these tumors (AK4.4) [121,122]. Consequently, losartan-treated tumors demonstrated an enhanced distribution and efficacy of oncolytic herpes simplex viruses as well as PEGylated liposomal doxorubicin (Doxil) [121]. In a phase II trial, losartan with FOLFIRINOX followed by radiotherapy resulted in a median overall survival of 33 months in patients with locally advanced PDAC [123]. Based on these promising results, a multi-institutional randomized phase II trial is currently underway (NCT03563248). Adenosine monophosphate-activated protein kinase (AMPK) also negatively regulates TGFβ signaling [124]. The pretreatment of subcutaneous PDAC (PANC-1 + PSC co-inoculated) mice with the AMPK activator metformin resulted in the decreased deposition of collagen I as well as fibroblast αSMA expression [125]. Furthermore, subcutaneous and orthotopic PDAC (PANC-1 + PSC co-inoculated) mice were treated with liposomes to deliver a nitric oxide donor (S-nitroso-N-acetylpenicillamine) to PSCs. This resulted in down-regulated TGFβ ligand expression and its downstream targets (collagen, fibronectin, and αSMA) [126]. In another example, halofuginone, which inhibits TGFβ signaling at the level of SMAD2/3 phosphorylation [127], attenuated PSC activation, decreased ECM deposition, and increased doxorubicin distribution in *KPC* mice [128].

The repurposing of therapeutic agents with indications in fibrotic conditions has also been tested. Pirfenidone is an oral anti-fibrotic used to treat idiopathic pulmonary fibrosis [129]. Pirfenidone treatment of subcutaneous and orthotopic PDAC (SUIT-2 + PSC co-inoculated) mice inhibited PSC proliferation and collagen deposition. Treatment with pirfenidone and gemcitabine together showed superior efficacy to either agent alone [130]. Another example is tranilast, which is a drug originally developed as an anti-allergic but currently also used in Eastern Asian countries to treat fibrotic conditions such as keloids and hypertrophic scars [131]. While direct evidence of efficacy in PDAC is scarce, the anti-fibrotic effects of tranilast in experimental models of breast cancer have been reported. Specifically, pre-treatment with tranilast decreased stromal collagen I and hyaluronan deposition as well as αSMA expression in fibroblasts, thus alleviating mechanical stress within the tumor of breast carcinoma-bearing mice [132,133]. Consequently, tranilast enhanced the delivery and efficacy of subsequently administered nanomedicines (e.g., nab-paclitaxel, Doxil, and docetaxel-loaded micelles). Notably, both pirfenidone and tranilast are known to modulate TGFβ signaling [131,134]. 

The stromal ablation strategies described above mainly targeted fibroblasts. Given the complex interplay of fibroblasts with various cell types including tumor cells and immune cells [40], stromal ablation strategies need not necessarily target fibroblasts per se. For example, mast cells generally promote fibrosis through the release of pro-fibrotic factors such as TGFβ, renin, chymase, and histamine [135] and also induce PSC proliferation in PDAC [136]. Ibrutinib/PCI-32765, an inhibitor of Bruton’s tyrosine kinase (BTK), is a potent systemic mast cell blocker. Ibrutinib dramatically reduced stromal collagen content in *KPC* mice and patient-derived xenografts, and the addition of ibrutinib improved survival compared to gemcitabine alone [137]. Based on these results, a phase III clinical trial assessed the benefit of adding ibrutinib to gemcitabine and nab-paclitaxel, but no improvement was seen in either overall survival or progression-free survival [138].

Importantly, caution is warranted when adopting stromal ablation as a therapeutic strategy. Numerous studies following the landmark study by Olive et al. [37] have revealed that stromal ablation ultimately results in disease progression and therapeutic resistance [139,140,141]. Notably, SHH inhibition in PDAC failed to improve patient outcomes in clinical trials [142]. As discussed in the following section(s), these studies have highlighted the complex role of fibrotic stroma in PDAC pathogenesis, progression, and therapeutic response. Stromal ablation also greatly affects tumor mechanics [101], which is increasingly appreciated to be a critical determinant of various aspects of PDAC biology and therapeutic outcomes [143,144].

### 3.2. Strategy #2: Stromal Reprogramming—Targeting Fibroblast Abnormalities

The failure of stromal ablation is generally understood to have revealed tumor-suppressive, in addition to the more well-studied tumor-promoting, roles of the fibrotic stroma in PDAC. This led to the *stromal reprogramming* strategy, which aims not to eradicate stromal components per se but to phenotypically alter the stroma to a more favorable state while retaining its tumor-suppressive functions (Figure 5). A well-known demonstration of this strategy targeted vitamin D receptor (VDR) to revert activated PSCs into a quiescent state [145]. The synthetic VDR agonist calcipotriol reduced stromal inflammation and fibrosis, increased intratumoral gemcitabine concentrations, and improved survival compared to gemcitabine alone in mice [145]. Nanoparticles co-delivering calcipotriol and SN38 decreased markers of fibrosis and improved therapeutic efficacy compared to SN38 alone or a mixture of calcipotriol and SN38 [146]. However, these encouraging pre-clinical studies are in contrast to the mixed results seen epidemiologically and clinically [147]. A recent in vitro study suggests that while calcipotriol favors the induction of tumor-suppressive fibroblasts, it may reduce T cell-mediated immunity [148]. Of note, a phase II pilot trial adding a VDR agonist (paricalcitol) and an immune checkpoint inhibitor (the anti-PD-1 monoclonal antibody nivolumab) to nab-paclitaxel, cisplatin, and gemcitabine has been conducted. While the effect of adding paricalcitol remains unclear due to the small sample size (10 patients) and lack of control groups, the promising overall response rate of 80% has led to an expansion of the trial to include 25 patients [149].

Another example of stromal reprogramming utilized all-*trans* retinoic acid (ATRA), which is an active metabolite of vitamin A. Quiescent PSCs are characterized by cytoplasmic lipid droplets storing vitamin A which become lost in activated PSCs [150,151]. Treatment with ATRA restored PSC quiescence, suppressed ECM remodeling, and slowed tumor progression in vitro and in *KPC* mice [152,153,154]. In humans, a phase I clinical trial assessing ATRA in PDAC revealed an increase in the apparent diffusion coefficient via diffusion-weighted magnetic resonance imaging [155], which is ascribed to the increased extracellular mobility of water molecules due to decreased collagen content [156]. 

The promising preclinical results of stromal reprogramming have motivated efforts to identify additional means of achieving stromal quiescence. A common strategy has been to identify genes up-regulated in fibrotic tissues of PDAC and/or chronic pancreatitis compared to healthy pancreata under the assumption that such genes are causally involved in fibrogenesis [157,158]. An example is OB-Cadherin/Cadherin-11, the genetic and/or pharmacological targeting (via SD133 treatment) of which attenuated markers of fibrosis and improved the therapeutic efficacy of gemcitabine in *KPC* mice [159]. Recent efforts to identify stromal reprogramming agents are increasingly fueled by a better understanding of CAF heterogeneity. The idea of CAFs as a heterogeneous population is not entirely new: early studies revealed differential marker protein (e.g., αSMA, vimentin, fibroblast specific protein 1, and NG2) expression across fibroblasts even within the same tumor [160]. Recent progress in culture techniques and single-cell analytics is accelerating the molecular characterization of the biologically and clinically relevant CAF subpopulations (Figure 6) [35,39,61,161,162].

By utilizing organoid co-culture models of PDAC tumor cells and PSCs, Öhlund et al. identified CAFs characterized by αSMA-positivity (named *myofibroblastic CAFs: myCAFs*) as well as another subpopulation characterized by the secretion of inflammatory mediators such as IL6 and a lack of αSMA expression (named *inflammatory CAFs: iCAFs*) [162]. myCAFs and iCAFs demonstrate differential localization within PDAC tissue: myCAFs exist adjacent to tumor cells, while iCAFs are located more distantly [162]. The existence of these subpopulations in PDAC has been replicated in multiple studies, although nomenclature varies [163,164,165,166,167]. A pan-cancer single-cell RNA sequencing study further suggests that these CAF subpopulations may be ubiquitous across cancer types [168]. The molecular mechanisms governing myCAF vs. iCAF differentiation are antagonistic, whereby the former is driven by TGFβ signaling which antagonizes IL1-induced JAK/STAT signaling that is important in the latter [115,165,169]. Moreover, hypoxia, which is characteristically observed in PDAC, strongly induced IL1 in tumor cells and favored iCAF differentiation [170]. The genetic ablation of STAT3 in PDAC CAFs in *KPC* mice slowed tumor progression, reduced collagen content, and alleviated immunosuppression [171]. The disruption of a myCAF phenotype via targeting Rho signaling promoted an iCAF phenotype [172]. SHH inhibition reportedly induced a myCAF-to-iCAF conversion, which could have also contributed to the clinical failure of SHH inhibition in PDAC [173]. A comprehensive single-cell RNA sequencing analysis has revealed another CAF subpopulation characterized by the expression of MHC class II and CD74 that activate CD4^+^ T cells in an antigen-specific manner (named *antigen-presenting CAFs: apCAFs*) [169]. Interestingly, apCAFs seem to be of mesothelial origin [73], again highlighting the heterogeneity of CAFs in PDAC.

A recent, notable step forward in our understanding of CAF heterogeneity is the identification of molecularly defined CAF subpopulations with tumor-suppressive functions (named *tumor-restraining CAFs* (*rCAFs*) in contrast to *tumor-promoting CAFs* (*pCAFs*)) [64]. The net balance of rCAFs vs. pCAFs has been suggested to determine the clinical outcome. Thus, the ultimate goal of stromal reprogramming is to efficiently shift the balance from “bad” stroma with an abundance of pCAFs in favor of “good” stroma characterized by rCAFs [75]. Mizutani et al. discovered that fibroblasts expressing the glycosylphosphatidylinositol-anchored membrane protein Meflin (also known as immunoglobulin superfamily containing leucine-rich repeat: ISLR) are tumor-suppressive in PDAC. Interestingly, they also showed that calcipotriol up-regulates Meflin expression, suggesting that stromal reprogramming via VDR agonism works in part through the induction of Meflin-positive rCAFs [174]. In a recent follow-up study, the synthetic retinoid Am80 was identified via chemical library screening to induce Meflin expression in CAFs. Am80 treatment attenuated tumor stiffening by impairing collagen cross-linking and alignment, and the addition of Am80 improved intratumoral drug delivery and therapeutic efficacy compared to gemcitabine alone in subcutaneous and orthotopic PDAC (mT5 inoculated) mice [175]. Based on these results, a phase I/II trial has been initiated to test the efficacy of Am80 in combination with nab-paclitaxel + gemcitabine [176].

### 3.3. Strategy #3: Targeting Fibroblast Metabolism

Cellular metabolism is increasingly understood as a key regulator of cell fate and function [177]. Deregulated cellular metabolism is a hallmark of cancer and a target for therapy [178,179,180]. The metabolic landscape of the TME is, in large part, shaped by disorganized tumor growth: consumption by proliferating tumor cells is believed to deprive the TME of nutrients. A comparison of human PDAC and adjacent benign tissue revealed that tumor tissues were low in glucose, upper glycolytic intermediates, as well as glutamine, serine, and creatine phosphate [181]. Another study compared nutrient levels in the TME (i.e., tumor interstitial fluid) vs. that in the circulation (i.e., serum) of *KPC* mice. The availability of nutrients such as glucose, tryptophan, arginine, and cystine was lower in PDAC TME compared to that in circulation [182]. Somewhat surprisingly, both studies revealed that levels of certain nutrients were elevated in tumor tissue, suggesting that nutrient availability is not entirely poor but complexly altered in the TME [181,182]. The altered nutrient availability in the TME poses unique demands on and forces the adaptation of the metabolism of constituent cells including fibroblasts [183,184]. The TME is also characteristically hypoxic and acidified, both of which also drive metabolic reprogramming [185,186]. 

Early studies on metabolic alterations in CAFs focused on increased aerobic glycolysis (dubbed the *reverse Warburg effect*) [187] and autophagy [188]. More recently, the nutrient-scarce PDAC TME was shown to drive macropinocytosis in CAFs via a calmodulin-kinase kinase 2 (CaMKK2)/AMPK-dependent pathway to support the survival of CAFs as well as tumor cells [189]. Cancer cells actively exploit and rely on the metabolism of CAFs to survive [188,189,190,191,192,193,194]. For a detailed discussion of metabolic alterations/reprogramming in the TME and how they pertain to PDAC pathogenesis and progression, the reader is referred to recent, dedicated reviews [180,183,184,195,196,197]. Our focus below will be to highlight some key findings on metabolic alterations in fibroblasts as it relates to fibrotic processes and the therapeutic implications of such alterations. Compared to our understanding of how the metabolism of CAFs is reprogrammed and exploited by tumor cells, the cell-autonomous consequences of metabolic alterations in CAFs are much less clear.

The metabolic/bioenergetic demands inherent to increased ECM synthesis and secretion in fibrosis may offer exploitable targets (Figure 7) [198]. ECM proteins are usually large, undergo various post-translational processing and modifications, and must be secreted. Energy is consumed throughout the whole process. At least partly due to such energetic (ATP) demands, heightened glycolytic activity is necessary for increased ECM production, myofibroblastic differentiation, and fibrosis progression [187,199,200]. A recent single-cell RNA sequencing study identified a CAF subpopulation (named *meCAFs*, for CAFs with an active metabolic state) characterized by heightened glycolytic activity and associated with a higher risk of metastasis, poor prognosis, but a better response to immunotherapy [167]. The up-regulation of various glycolytic enzymes is often observed in cancer, and these enzymes have garnered interest as potential therapeutic targets [201,202]. Autophagy is also important in PSC activation and the secretion of ECM molecules as well as inflammatory cytokines such as IL6 [190]. Similar to glycolysis, autophagy too can also be targeted at various points of its regulatory pathways [203]. 

Collagen is a collective term encompassing a large family of glycoproteins characterized by (1) a repetition of the amino acid sequence [Gly-X-Y]_n_, (2) the occupation of X and Y positions by proline and hydroxyproline, and (3) the formation of a right-handed triple helix from three left-handed polyproline α-chains of identical length (for a comprehensive review of collagens, the reader is referred to [77,205]). Collagens consist of over 90% of the total ECM mass throughout PDAC progression, even while the total ECM mass increases over 2-fold in PDAC compared to normal pancreata [78]. The demanding biosynthetic requirements for fibroblasts are further compounded by the unique amino acid composition (approximately 30% glycine and 15–20% proline or hydroxyproline) of collagens [198]. The idea of targeting this process to ameliorate fibrosis is not entirely new [206]. In particular, the requirement of proline in collagen synthesis seems to be a viable therapeutic target. Autophagy deficiency led to decreased collagen synthesis, which is in part due to the role of autophagy in promoting proline biosynthesis [207]. More recently, pyrroline-5-carboxylate reductase 1 (PYCR1), a key enzyme in proline biosynthesis, in CAFs was shown to be necessary for collagen production in breast cancer [208]. This vulnerability seems to be because fibroblasts tend to synthesize their own proline unless precursor metabolites (such as glutamine) are limiting [198]. To date, the effect of targeting proline biosynthesis on nanomedicine delivery in PDAC is unknown but should be assessed in future studies.

Another aspect of CAF metabolism that warrants attention is its altered redox status. Oxidative stress is a common feature of various fibrotic conditions, and it seems to play an important role in fibrogenesis. The pro-fibrotic TGFβ induces ROS production via various mechanisms such as the induction of ROS-producing NADPH oxidases as well as suppression of antioxidant enzymes such as superoxide dismutase. Indeed, NADPH oxidase is important for PSC activation [209]. Increased ROS levels, in turn, play a role in TGFβ activation from its latent form, forming a vicious loop [210,211]. Interestingly, TGFβ promotes proline biosynthesis in a SMAD4-dependent fashion in fibroblasts concurrently with increased glucose and glutamine uptake to meet the bioenergetic demands of ECM production. The induction of proline biosynthesis acts as a redox vent to utilize excessive mitochondrial redox potential, thereby protecting fibroblasts from oxidative damage [212]. Increased ROS inactivates Kelch-like ECH-associated protein 1 (KEAP1), which is responsible for the degradation of the transcription factor NF-E2-related factor 2 (NRF2) under unstressed conditions. ROS-induced activation of NRF2 up-regulates various detoxifying enzymes to counteract oxidative stress [213]. The deletion of *Nrf2* in *KPC* mice and PDAC organoids revealed a pro-tumorigenic role of NRF2 in PDAC [214,215]. Further study demonstrated that Nrf2 expression in PSCs was sufficient to promote tumorigenesis [216]. Of note, *Nrf2* deletion reduced the stromal reaction in *KPC* mice [215]. The above results suggest that the redox metabolism is a potential reservoir of therapeutic targets to counter fibrosis.

With an increasing understanding of metabolic alterations in cancer and therapeutic agents to target them [217], it is also important to note that enzymes and associated regulatory proteins of a particular metabolic process usually also possess functions outside of this process [218]. The pleiotropic function of metabolic enzymes/regulatory proteins must be kept in mind during the interpretation of experimental results and therapeutic implementation.

### 3.4. Strategy #4: Targeting ECM Abnormalities

ECM deregulation in the fibrotic TME leads to tumor stiffening (i.e., heightened Young’s modulus), high mechanical stress, and high interstitial fluid pressure (IFP), all of which limit the intratumoral delivery of nanomedicine [100,219,220,221,222,223]. Therefore, an important goal of targeting ECM abnormalities is to normalize tumor mechanics [143,144]. As we discuss in detail below, strategies targeting ECM abnormalities can aim to therapeutically degrade ECM components aberrantly deposited (strategy #4–1), target the aberrant ECM remodeling (strategy #4–2), and/or target the signaling induced by the aberrant ECM (strategy #4–3). Although much remains to be studied, we highlight some key findings/concepts and discuss future directions below.

#### 3.4.1. Strategy #4–1: Therapeutic Enzymatic Degradation of ECM Components

ECM components in the fibrotic PDAC TME hinder the passage of nanomedicine via various mechanisms (Figure 8) [221,222]. Fibrillar collagens sterically block the passage of nanomedicine [224]. ECM components are compressed into a dense and tortuous network due to the solid stress posed chiefly by increased cellularity, and tortuosity affects the effective distance nanomedicines must travel [225]. ECM cross-linking induces stiffening, also contributing to elevated solid stress [226]. Some ECM components, especially the glycosaminoglycans, trap water and decrease hydraulic conductivity (i.e., the ease with which water can percolate through the extracellular milieu) [227]. Increased ECM deposition results in heightened IFP, which also reduces the penetration of nanoparticles into the tissue [220]. Enzymatic degradation of ECM components has been attempted as a therapeutic strategy to overcome these issues.

Hyaluronan is a glycosaminoglycan consisting of disaccharide repeats of N-acetylglucosamine and glucuronic acid. Its molecular weight varies between 100 and 10,000 kDa, which seems to have functional consequences [228,229]. Hyaluronan is ubiquitously present throughout the human body but overproduced in many types of human cancers, including PDAC [229,230]. Hyaluronan participates in cellular signaling via receptors such as CD44 and RHAMM [228,229]. Hyaluronan is also a nutrient fueling cancer cell proliferation in PDAC [231]. Apart from these biological functions, hyaluronan binds/traps large amounts of water due to its negative charge [227]. This ensures tissue hydration under physiological conditions but decreases hydraulic conductivity and increases IFP in pathological conditions such as cancer [227,228]. In *KC* (*LSL-Kras^G12D/+^; Pdx1-Cre*) and *KPC* mice, hyaluronan accumulation was observed early in PDAC tumorigenesis [232]. Overproduced hyaluronan increased IFP and impaired vascular function, resulting in poor drug delivery [232,233,234]. This was reversed by treatment with pegvorhyaluronidase alfa (PEGPH20), which is a PEGylated recombinant human hyaluronidase [232,233,234]. Based on these pre-clinical results, PEGPH20 was tested clinically. However, a phase IB/II randomized study of PEGPH20 with FOLFIRINOX in patients with metastatic PDAC demonstrated that the addition of PEGPH20 is detrimental in patients unselected for tumor hyaluronan status [235]. Even after selecting for patients with high hyaluronan levels, a randomized phase III trial of PEGPH20 with nab-paclitaxel plus gemcitabine in metastatic PDAC revealed that PEGPH20 significantly increased overall response rate but failed to improve progression-free survival or overall survival [236]. The clinical development of PEGPH20 in PDAC has thus been discontinued. As with the clinical failure of stromal ablation, it has been suggested that the hyaluronan-rich stroma may also possess tumor-suppressive roles [237]. 

Another ECM component targeted for enzymatic degradation is collagen. Collagens are major constituents of the ECM, and its abnormal deposition is a hallmark feature of fibrosis as well as in PDAC [42,78]. Fibroblasts up-regulate collagen expression in response to interacting with cancer cells, which ultimately results in decreased nanomedicine passage [238]. Especially, the fibrillar collagens types I and III are overproduced [239]. Collagens, and their proteolytic fragments, elicit intracellular signaling via cellular receptors [240,241]. Moreover, collagens, due to their abundance, also serve as a nutrient source for cancer cells in PDAC [242]. Overproduced collagens become increasingly linearized during PDAC tumorigenesis/progression in *KC* and *KPC* mice and are also clinically associated with worse prognosis [101,243]. The deposition and linearization of collagens result in tumor stiffening, which impedes drug penetration and further promotes pro-fibrotic signaling pathways [243,244]. The enzymatic degradation of collagen improved the nanomedicine delivery of paclitaxel micelles in an orthotopic PDAC (*KPC* cell-inoculated) mice model of PDAC [245]. Of note, genetically deleting collagen I expression in αSMA-positive myofibroblasts accelerated PDAC in *KPC* mice, indicating tumor-suppressive roles of collagen I [246]. “Good” and “bad” collagens thus exist, but the role of collagens in pathogenesis seems highly context-dependent [205]. Further studies are warranted, but caution is necessary when therapeutically implementing collagen degradation.

#### 3.4.2. Strategy #4–2: Targeting ECM Remodeling

Abnormalities in ECM remodeling—ECM deposition/post-translational modification, force-mediated ECM modification, and ECM degradation—are characteristically observed in cancer [81,83,84,86,87] and offer various opportunities for therapeutic exploitation.

*ECM deposition/post-translational modification*. An obvious target during this step would be to decrease ECM synthesis. This is, in large part, a goal pursued by stromal reprogramming. In the case of collagen, the inhibition of fibril formation might be a therapeutic target in fibrosis. Antibodies directed against the telopeptide region of collagens [247] as well as peptides blocking key interactions between procollagen chains [248] interfere with collagen fibril formation. Interestingly, the composition of the collagen triple helix is altered in PDAC: the collagen I triple helix normally consists of two α1 chains and one α2 chain, but in PDAC, it consists of three α1 chains due to the epigenetic suppression of *Col1a2* (the α2 chain-encoding gene). The altered composition enhances cancer cell proliferation via integrin α3β1 signaling and suppresses anti-tumor immunity [249]. Altered composition offers an opportunity for tumor-specific therapeutic intervention and warrants further study. Whether other collagens also demonstrate altered composition is unknown. Chaperones dedicated to collagens such as heat-shock protein 47 (HSP47)/Serpin H1 are also potential targets [250]. HSP47 is up-regulated in PDAC stroma [158,251]. Targeting HSP47 with siRNAs in PSCs decreased the collagen content and ameliorated pancreatic fibrosis in mice [252,253]. Small molecules targeting HSP47 have also been identified [254,255], but whether it is efficacious against fibrotic barriers in PDAC remains to be studied. Cross-linking is an ECM post-translational modification that has been extensively studied as it drives tumor stiffness and induces mechanosignaling [256]. Collagen cross-linking is driven by enzymes such as lysyl oxidase (LOX) and transglutaminase [83]. *LOX* expression is elevated in *KPC* mice and is clinically associated with a worse prognosis in PDAC [257]. Tumor tissues of *KPC* mice are stiffer (Young’s modulus of 11.3 ± 1.7 kPa) than normal pancreata (4.2 ± 1.3 kPa). Human PDAC tissues (44.8 ± 5 kPa) also are stiffer than neighboring non-neoplastic pancreata (3.5 ± 0.5 kPa) [258]. Targeting LOX with an antibody or with the small molecule inhibitor beta-aminopropionitrile decreased fibrillar collagen and softened tumor tissue [243,257]. Importantly, the addition of an anti-LOX antibody to gemcitabine demonstrated therapeutic efficacy superior to either treatment alone in *KPC* mice [257]. Another collagen cross-linker, transglutaminase-2, is secreted by PDAC cells and is enzymatically active in PDAC stroma. Orthotopic PDAC (AsPC1 or PANC-1 inoculated) mice in which transglutaminase-2 was knocked down in tumor cells demonstrated significantly thinner collagen fibers as well as decreased collagen content and reduced CAF activation [259]. Especially with the development of transglutaminase-2 inhibitors [260], future studies assessing whether targeting transglutaminase-2 in a therapeutic setting improves drug delivery are warranted.

*Force-mediated ECM modification*. This step involves the engagement of ECM molecules by cellular receptors [83]. ECM receptors such as integrins transduce forces generated by cells to the ECM, which drive ECM organization at the tissue scale [97,261]. While collagen fibers are randomly oriented in healthy tissue, anisotropy (high degree of alignment) is often observed in PDAC and has been reported as a negative prognostic factor following surgical resection [262,263]. PSCs from PDAC demonstrated an increased ability to align ECM fibers compared to normal fibroblasts in part via a TGFβ/ROCK signaling-dependent mechanism [36]. ECM alignment allows the coordination of cellular forces over large areas. Compounded with the acquisition of myofibroblastic contractility by CAFs, this could result in a further contraction/stiffening of tumor tissue [264]. Force-mediated ECM modification seems to work in parallel with ECM post-translational modifications, as the inhibition of cross-linking enzymes such as LOX disrupted ECM alignment [243,265]. Indeed, pre-treatment of orthotopic PDAC (*KPC* + PSC co-inoculated) mice with a lysyl oxidase-like 2 (LOXL2) inhibitor disrupted collagen alignment without altering content, softened tumor tissue, and improved the therapeutic efficacy of nab-paclitaxel [265]. The force applied by cells onto ECM molecules can also cause intramolecular deformations that reveal cryptic binding sites, which could affect ECM signaling [266,267]. One of the best-studied examples is fibronectin, in which individual domains within the molecule change conformational status (i.e., folded vs. unfolded) in response to applied force [268,269,270]. Peptide probes have been generated that discriminate force-induced conformational states of fibronectin, and probes specific to relaxed fibronectin showed preferential accumulation and retention in tumor tissue compared to other organs [271,272]. 

*ECM degradation*. This step is driven by target-specific proteases such as the various MMPs, disintegrin and metalloproteinases (ADAMs), disintegrin and metalloproteinases with thrombospondin motifs (ADAMTSs), as well as serine, cysteine, aspartate, and threonine proteases [273]. ECM degradation must be tightly coordinated with ECM synthesis, modification, and organization to achieve homeostasis, but it is deregulated in cancer [80,83,273,274]. One might expect ECM degradation to be decreased in fibrosis, but this is not the case [275]. ECM degradation drives ECM turnover in which the normal ECM is replaced by tumor-derived ECM during tumorigenesis and fibrogenesis [83]. Indeed, the broad inhibition of MMPs with GM6001 attenuated the aberrant ECM remodeling by PSCs from PDAC [36]. Importantly, ECM degradation is not a solely structural phenomenon. ECM degradation provides nutrients for cancer cells and initiates cellular signaling via bioactive fragments generated during degradation [241,242]. For example, collagen I, which while intact demonstrates tumor-suppressive functions, signals through the discoidin domain receptor (DDR) 1–NFκB–p62–NRF2 axis to promote PDAC growth following MMP-dependent collagenolysis [276]. ECM-bound growth factors are also released upon ECM degradation [277]. Post-translational modifications can alter ECM degradation dynamics. Indeed, cross-linked and bundled collagen fibers found in fibrosis are resistant to degradation [278]. A recent systematic review, however, found that the available evidence regarding the role of MMPs in PDAC progression is quite inconsistent [279]. This perhaps reflects the highly divergent and context-dependent roles of the various MMPs [275,280]. MMPs and other enzymes implicated in ECM degradation also have non-ECM substrates and at times even work in a nonproteolytic manner [82,275,281]. While ECM-degrading enzymes in the fibrotic stroma have been exploited as a TME-specific on-switch for nanomedicine delivery (detailed in Section 4.3), the effect of targeting ECM-degrading enzymes on fibrotic barriers in PDAC remains largely unstudied. Caution is warranted, however, as initial experience with MMP inhibitors in the clinic was poor with unexpected, severe side effects [280].

#### 3.4.3. Strategy #4–3: Targeting ECM Signaling

ECM signaling is deregulated in fibrosis as well as in cancer and offers various therapeutic opportunities [282]. The importance of integrins in CAF differentiation and function was recently reviewed elsewhere [98]. In PDAC in particular, integrins β1 [283], α3 [284], α5 [285,286], αV [287,288], and α11 [289] in PSCs/CAFs have been hitherto targeted and studied. The most promiscuously interacting subunit, integrin β1, signals through focal adhesion kinase (FAK) in PSCs to radioprotect PDAC cancer cells [283]. PDAC CAFs express integrin α3 more abundantly than normal fibroblasts in vitro, and α3 knockout impaired CAF differentiation [284]. Integrin α5 is up-regulated in the fibrotic stroma of PDAC compared to healthy pancreatic tissue and is clinically associated with decreased survival. Knockdown of *ITGA5* (encoding integrin α5) inhibited myCAF differentiation in vitro via its effect on TGFβ/SMAD as well as FAK signaling [285]. Interestingly, not only expression but intracellular localization seem to be important: the redistribution of active integrin α5β1 from the cell surface to assorted endosomes was found to drive PDAC CAF differentiation, and patients with integrin α5β1 localized mainly at the cell surface had better prognosis [286]. Integrin αV/CD51 is highly expressed in PDAC PSCs, and stromal expression of integrin αV was associated with advanced disease and decreased survival. Knockdown of *ITGAV* reduced PSC proliferation and migration [288]. Integrin αVβ5 was also necessary for the endosomal redistribution of integrin α5β1, pointing to the importance of crosstalk between the various integrins [286]. Integrin α11 expression in PDAC stroma is higher than in adjacent non-tumoral tissues, and *ITGA11* knockdown impaired PSC activation [289]. Treatment with a peptidomimetic inhibitor of integrin α5 (AV3) or a designed protein inhibitor of integrin αVβ3 (ProAgio) in PSCs ameliorated fibrosis, decreased tumor growth, and enhanced sensitivity to gemcitabine in mice (PANC-1 + PSC co-inoculated mice in [285] and *KPC* mice in [287]). Integrin inhibitors have long been pursued in various clinical applications including fibrotic conditions [290,291]. Despite setbacks, increased biological and structural insights into integrin biology are fueling the development of novel inhibitors [291,292]. 

An important aspect of integrin-mediated ECM signaling is its role in mechanotransduction [92,261]. The general role of mechanotransduction in fibrosis has recently been reviewed [293,294,295]. Mechanical forces regulate ECM-integrin binding kinetics as well as the activation/conformation, clustering, and trafficking of integrins [92]. The compositional dynamics of the integrin adhesome are also mechanically regulated [296]. The combined biochemical and mechanical properties of integrins enable the integration of various microenvironmental signals which then signal through various downstream effectors [92,266]. These effectors could also offer various therapeutic opportunities. Signaling downstream of integrins is centered around four core pathways: ILK–PINCH–kindlin, FAK–paxillin, talin–vinculin, and α–actinin–zyxin–VASP pathways [297]. Of these pathways/components, ILK [298], kindlin-2 [299], and FAK [300] have been reported to be important for PSC/PDAC CAF activation. Moreover, VASP is important for the activation of the closely related hepatic stellate cell in hepatic fibrosis via the promotion of TGFβ signaling [301]. Additionally, the stromal expression and/or activation of PINCH [302], kindlin-2 [299], and/or FAK [300] have been associated with poor clinical outcomes in PDAC. Indeed, pharmacological inhibition of FAK (e.g., VS-4718 and PF-562271) reduced ECM expression and fibrosis in *KPC* mice [300,303]. FAK inhibition rendered *KPC* mice more sensitive to gemcitabine and nab-paclitaxel treatment in part through tumor softening [304]. VS-4718 treatment in *KTC* mice (*LSL-Kras^G12D/+^; Tgfbr2^flox/+^*; *Ptf1a-Cre*) decreased stromal content and augmented the efficacy of a combination treatment consisting of G47Δ (a third-generation oncolytic herpes simplex virus type 1) and immune checkpoint inhibition [305].

A common cellular process downstream of integrin signaling is the remodeling of the actin cytoskeleton [306]. Actin remodeling is crucial for the acquisition of a myCAF phenotype, as can be observed by the acquisition of the expression of specific actin isoforms upon activation (i.e., αSMA) [307,308]. For example, ROCK is a critical regulator of the actin cytoskeleton, and its inhibition attenuated PSC activation in vitro [309]. The expression of both ROCK1 and ROCK2 isoforms increases with tumor progression in murine and human PDAC, and elevated ROCK expression portends a poor prognosis [310]. ROCK promotes collagen remodeling, and treatment with various ROCK inhibitors (e.g., Fasudil, dimethylfasudil/H1152, Y27632, and AT13148) prolonged survival in *KPC* mice [310,311]. Interestingly, ROCK inhibition need not be chronic, and transient treatment seems to suffice. In a strategy dubbed *tissue priming*, transient ROCK inhibition with Fasudil attenuated fibrotic remodeling and improved sensitivity to treatment with gemcitabine and nab-paclitaxel [312]. In addition to ROCK, the actin-organizing protein palladin was recently reported to be important in fibroblast activation in response to TGFβ as well as for the generation of tumor-supporting ECM [313]. Indeed, stromal palladin expression is a negative prognostic factor in PDAC [314].

Non-integrin ECM receptors such as DDRs also play important roles in fibrosis [315,316]. The two DDRs, DDR1 and DDR2, are RTKs that bind collagens [240,316,317]. DDR1 and DDR2 do not seem to be functionally redundant and might even have opposing roles: DDR1 is pro-fibrotic while DDR2 is anti-fibrotic [315]. The inhibition of DDRs with imatinib reduced PSC activation and ECM deposition in cerulein-induced pancreatitis [318]; the caveat is that imatinib is not selective to either DDR1 or DDR2 and also inhibits a variety of other RTKs. Moreover, *DDR1* knockout in *KPC* mice resulted in pancreatic atrophy accompanied by increased fibrillary collagen deposition [319], which might be problematic when targeting DDR1 in PDAC.

## 4. Therapeutic Strategies Optimizing Nanomedicine Design to Overcome Fibrotic Barriers to Nanomedicine

The focus so far has been on therapeutically modulating the fibrotic TME (strategies #1–#4). As mentioned above, the optimization of nanomedicine design will likely also be important in overcoming the fibrotic barriers in PDAC. Various avenues of optimization exist, but we envisage the following four categories (Figure 4): physicochemical optimization of nanomedicines (strategy #5), installation of active targeting properties (strategy #6) and/or microenvironmental responsivity (strategy #7) in nanomedicines, as well as the design of nanomedicine as nanosensitizers to physically manipulate the fibrotic stroma (strategy #8). In the following sections, we discuss general considerations in nanomedicine design by focusing on the fibrotic stroma without necessarily delving into the technical details of various nanomedicine formulations. Although we specifically focus on fibrotic barriers in this review, other biological barriers must also be kept in mind during nanomedicine design. Optimization must be executed comprehensively to achieve maximal delivery and efficacy throughout the biodistribution of the administered nanomedicine. It is also of note that many issues must be addressed for the successful clinical translation of nanomedicine [320,321,322,323]. We emphasize the need for formulations to be suited for mass production: their synthesis must be controllable, reproducible, and scalable [321]. Generally speaking, the more complex/sophisticated a formulation becomes, the more difficult it is to mass produce; this tradeoff must be kept in mind during the optimization of nanomedicine design.

### 4.1. Strategy #5: Optimizing the Physicochemical Properties of Nanomedicine

Less than 1% of injected nanomedicines accumulate in tumors, and only about 0.001% eventually interact with cancer cells due to the presence of various biological barriers [24,324]. Improving intratumoral accumulation and delivery to cancer cells is thus of paramount importance [24,321,323,324]. Conceptually, nanomedicine formulations consist of payload (i.e., the active drug to be delivered) and non-payload (i.e., scaffold to integrate the payload and necessary functional moieties into a unitary agent) portions. The non-payload portion, often comprising the majority of the nanomedicine formulation, is not inert. Indeed, numerous physicochemical parameters of nanomedicine—irrespective of deliberate functionalization—are known to elicit biological effects (called the *ancillary effects* of nanomedicine formulations) and affect the efficiency of delivery [325]. For improved delivery, these parameters must be optimized (Figure 9). Examples include particle size, particle geometry, surface charge (zeta potential), surface chemistry, and elasticity, among various others [24,321,324].

Particle size greatly affects the ability of nanomedicine to penetrate fibrotic barriers in PDAC. A comparison of platinum-loaded micelles of various diameters (30, 50, 70, and 100 nm) revealed that only 30 nm micelles could penetrate the fibrotic stroma sufficiently to achieve a therapeutic effect in subcutaneous PDAC (BxPC-3 inoculated) mice [326]. The small nanoparticle size poses an engineering challenge especially when the payload is of large molecular weight, as in the case of nucleic acid therapeutics. However, strategies to deliver nucleic acid therapeutics in the <30 nm size range have already been devised. For example, dynamic interactions between Y-shaped block catiomers of precisely controlled chain lengths with siRNAs have been utilized to generate unit polyion complexes (uPICs) with a size of ~18 nm [327]. Another strategy complexed single-stranded DNA (instead of double-stranded DNA that is bulkier due to its structural rigidity) with poly(ethylene glycol)-*b*-poly(l-lysine) to generate polyplex micelles with a size of ~29 nm [328]. The successful penetration of fibrotic stroma and the therapeutic effect of both nanomedicine systems have been tested in subcutaneous PDAC (BxPC-3 inoculated) mice [327,328]. The dependence of nanomedicine delivery efficiency and therapeutic efficacy on particle size is likely a result of various filtration processes. While it is not definitively clear what confers size selectivity within the fibrotic stroma, ECM density and composition are likely important [329]. Interestingly, size selectivity seems to be different between models [330]. This raises the possibility of inter-patient variability in optimal nanomedicine particle size, which could be problematic for clinical translation [331]. The efficiency of retention must also be considered: while smaller particles more easily permeate, larger particles are better retained within tissues. Various strategies have thus been proposed to achieve size modulation within single nanoparticles in response to various external stimuli [324,332].

The effects of various physicochemical properties on cellular uptake have been intensively studied [333,334,335]. This could inform the design of nanomedicine that evades unintended uptake and clearance by phagocytic cells, such as macrophages, which are abundant in the fibrotic PDAC TME [336,337,338]. Chemical modification of the nanoparticle surface by PEGylation is a safe and clinically utilized method to avoid macrophage uptake [339]. On the other hand, given the essential role of macrophages in tissue remodeling and fibrosis [336,337,340], targeting macrophages by exploiting the innate tendency of these cells to uptake nanomedicine may also be leveraged for therapy [338]. Such a strategy could aim to reduce macrophage content altogether or attempt to shift macrophage polarization away from a fibrosis-promoting phenotype and induce a fibrosis-resolving phenotype [336,341].

The effects of physicochemical parameters other than particle size on the penetration efficiency of fibrotic barriers are relatively under-studied in contrast to cellular uptake. Regarding particle geometry (shape), a study comparing spherical vs. worm-like nanoparticles of similar particle size (~110 nm) as determined by dynamic light scattering reported enhanced accumulation of the latter in subcutaneous PDAC (B33 inoculated) mice. A similar trend was observed but not statistically significant in *KPC* mice [330]. Various non-spherical nanoparticles have been hitherto devised [342,343], and a comprehensive comparison of different particle geometries and their effect on the penetration of the fibrotic barriers in PDAC is warranted. In regard to the surface charge, positively charged nanoparticles showed somewhat lower tumor accumulation than their neutral/anionic counterparts of similar size (~100 nm) but demonstrated improved penetration and antitumor efficacy in subcutaneous PDAC (BxPC-3 inoculated) mice [344]. As illustrated here, the surface charge can sometimes have opposing effects on different phases in the process of nanomedicine biodistribution. Strategies to modulate surface charge within single nanoparticles in response to various external stimuli have thus been devised [324,345]. Particle elasticity [346] and stiffness [347] also affected tumor accumulation in models of breast and ovarian cancer, respectively. However, because these reports did not specifically assess fibrosis, the effect of these parameters on the penetration efficiency of fibrotic barriers in PDAC remains unclear.

### 4.2. Strategy #6: Active Targeting of Nanomedicine

The delivery of nanomedicine is driven by diffusion (movement governed by concentration gradient) and convection (movement governed by fluid flow) [222,324]. *Passive targeting* solely relies on these forces, whereas *active targeting* strategies involve the installation of moieties that exploit features (such as the specific up-regulation of a particular protein) characteristic of the TME onto the surface of the nanomedicine formulation (Figure 10) [348]. Strategies for *subcellular targeting* (or *organellar targeting*), the targeted delivery of the payload (to specific organelles) after cellular internalization, is also an active field of research [349,350]. Indeed, to achieve the maximal active targeting effect, a hierarchical strategy that achieves tissue-type, cell-type, and intracellular specificity is necessary [350]. Moieties utilized in active targeting include small molecules, polypeptides of various lengths (small peptides to proteins such as antibodies), and aptamers [351,352,353]. Active targeting may also build upon the specificity of natural vectors, such as EVs and viruses [354,355]. Active targeting was initially proposed to increase localization to and retention at target tissues as well as enhance uptake by target cells [331,351]. There is some debate as to how active targeting functions, but it seems that it does not necessarily work by significantly increasing overall tumor localization but rather by enhancing cellular uptake [24,331]. Interestingly, active targeting using the V7 peptide (a pH-low insertion peptide: pHLIP) which exploits the acidic TME outperformed particle size modulation in orthotopic PDAC (S2VP10 inoculated) mice [356], with the caveat that the histological evaluation of fibrosis was not performed in the study.

Active targeting generally is directed at cellular targets, especially proteins over-expressed on the surface of cancer cells [352,357]. Strategies to target stromal cell types are much less explored except for tumor endothelial cells that have been extensively targeted given their role in regulating the extravasation of nanomedicine into the tumor tissue [348,358]. Attempts have been made to target fibroblast activation protein (FAP), which is a membrane protein up-regulated in PDAC CAFs. The stromal expression of FAP in PDAC correlates with desmoplasia and worse prognosis clinically [359,360]. FAP^+^ CAFs mediate immunosuppression in PDAC via the secretion of CXCL12 [361]. However, targeting FAP^+^ CAFs showed little therapeutic benefit and further resulted in cachexia and anemia [362,363]. The latter adverse effects were shown to be due to collateral damage against FAP-expressing multipotent bone marrow stem cells [363]. This example illustrates that specificity is especially crucial, and the unintended targeting of even numerically minor cell populations can have dire, systemic consequences. The active targeting of fibroblasts thus requires the identification of highly specific, membrane-expressed markers, which could be complicated by the highly heterogeneous nature of CAFs.

In addition to directly targeting fibroblasts, the abundantly deposited ECM in fibrosis can also be targeted [364]. Compared to simply targeting the cancer cells, targeting the ECM within the TME might have the additional benefit of affecting tumor-supportive stromal cells. An example is the cancer-stroma targeting (CAST) therapy [365]. In the CAST therapy, cytotoxic agents are conjugated to ECM-targeting antibodies (which result in an antibody–drug conjugate [ADC]). The ECM-bound ADCs serve as a scaffold from which cytotoxic agents are sustainably released and diffuse through the tumor tissue. An ADC consisting of SN38 conjugated to an anti-collagen IV monoclonal antibody showed therapeutic efficacy against subcutaneous PDAC xenograft (SUIT-2 or PSM1 inoculated) mice [366]. ECM-directed nanomedicines do not necessarily need to be ADCs, and other nanoparticles can also in principle be actively targeted to the ECM. 

Apart from collagen IV, other ECM components such as tenascin-C, fibronectin, aggrecan, heparan sulfate, and chondroitin sulfate have been targeted in other cancer types [364]. Given the strong expression of tenascin-C and fibronectin in the PDAC stroma compared to healthy tissue, these ECM components may be promising targets but remain unassessed [367]. Connective tissue growth factor (CTGF) is chiefly produced by PSCs and significantly over-expressed (>40-fold induction over normal tissue) in PDAC [368,369]. CTGF expression level correlates with the degree of fibrosis [368], and thus, CTGF could potentially be actively targeted. Interestingly, targeting CTGF with the monoclonal antibody FG-3019 improved the efficacy of gemcitabine in *KPC* mice. However, intratumoral gemcitabine concentration was not affected by FG-3019 treatment, pointing at mechanisms independent of drug delivery, at least of small molecular weight cytotoxic agents [370]. Secreted protein acidic and rich in cysteine (SPARC) is a matricellular protein whose expression in CAFs is associated with an activated stroma phenotype and portends a poor prognosis [371,372]. SPARC expression is driven by the interaction of CAFs with PDAC cancer cells [373]. Thus, SPARC could potentially be actively targeted. Of note, stromal SPARC is necessary for collagen deposition in PDAC [36,374]. Although *Sparc* knockout in *KC* mice reportedly did not lead to increased intratumoral gemcitabine concentrations [374], its effect on nanomedicine delivery remains unknown. In line with reports that the genetic ablation of stromal collagen I reduces survival in *KPC* mice, a profound reduction in survival was seen also upon *SPARC* knockout in *KC* mice as would be expected from its effect on decreased intratumoral collagen content [246,374]. Interestingly, SPARC was initially thought to augment the efficacy of nab-paclitaxel due to its albumin-binding potential [375], although confirmation studies have offered conflicting results [376,377]. 

Proteomic analysis of the PDAC stroma [78,378] could guide the identification of over-expressed ECM targets for active targeting, although tumor specificity warrants systemic validation. In addition to over-expressed ECM components, active targeting may also be directed at abnormally remodeled ECM components. For example, peptide probes discriminating between different tensional/conformational states of fibronectin, which result from force-mediated ECM modification, have been used for imaging prostate cancer xenografts in mice [183]. Peptide probes that bind fibrillary collagens with damaged triple-helix structures have been utilized to visualize areas of fibrosis and inflammation [280,281]. Currently, the utility of these probes for therapeutic or theranostic purposes in PDAC is unknown, and future studies are warranted.

Whatever the strategy employed for active targeting, it must be kept in mind that nanomedicines interact with the surrounding biological microenvironment once administered, and these interactions lead to various ancillary effects [325]. A notable example is the rapid formation of a protein corona on the surface of nanomedicines in the blood, which results from the interaction between the nanomedicine and serum proteins [379,380,381]. Protein corona formation is nanoparticle-specific and dynamically evolves [382,383,384]. Protein coronas may hamper the functionality of targeting moieties [385]. On the other hand, protein corona formation can be engineered to design “stealth” nanomedicines that evade clearance by the mononuclear phagocytic system [380,381].

### 4.3. Strategy #7: Installing Microenvironmental Responsivity in Nanomedicine

The PDAC TME poses a unique milieu distinct from that of normal pancreata [386], the characteristics of which could serve as a microenvironmental on-switch for nanomedicines. The installation of microenvironmental responsivity in nanomedicine would allow spatiotemporal control of its therapeutic effects (Figure 11A,B) [387,388,389,390]. Especially in the case of polymeric micelles, an ever-expanding body of work has focused on designing “smart” polymers incorporating moieties that confer responsivity to stimuli such as a change in pH and redox status as well as the presence of various enzymes, metabolites, or ROS [391]. We refrain from delving into the specific chemistries which enable each stimuli-responsivities (the reader is referred to dedicated reviews elsewhere: [391,392,393,394,395,396,397,398,399,400]) and focus on characteristics of the PDAC TME driven by fibrosis that may be exploited for therapy. 

Hypoxia and extracellular acidity are characteristics of PDAC. Fibrosis plays an important role in the generation of both. PDAC is known for its hypovascularity [37]. Fibrosis seems to inhibit angiogenesis and decrease vessel density [140], although in vitro studies report that PSCs produce angiogenic growth factors [55]. Increased IFP due to hyaluronan accumulation in the fibrotic stroma leads to vascular compression/collapse, which impedes vascular perfusion [232]. Hypovascularity compounded by decreased perfusion leads to hypoxia in PDAC [401]. A study of seven PDAC patients revealed a markedly decreased median tissue partial oxygen pressure (range: 0–5.3 mmHg) compared to the adjacent normal pancreas (24.3–92.7 mmHg) [402]. Hypoxia, as well as the associated metabolic reprogramming characterized by enhanced glycolysis in cancer cells and CAFs alike, leads to an acidic extracellular milieu. Using acido-chemical exchange saturation transfer (acidoCEST) magnetic resonance imaging, it was shown that *KC* mice treated with cerulein to induce PDAC demonstrated lower extracellular pH (~6.75) compared to normal mice or *KC* mice without cerulein treatment (6.92~7.05) [403]. Consistent results have been obtained using pHLIPs which activate at pH < 6.8. Orthotopic PDAC (S2VP10, S2013, or Capan-2 inoculated) mice and *KPC* mice demonstrated an enhancement of pHLIP accumulation suggesting low extracellular pH, although the conversion to exact pH values is difficult [404,405]. As we mentioned above, pHLIPs have been previously utilized for active targeting in experimental PDAC [356]. At least theoretically, the tumor core is expected to be more acidic than the tumor periphery. Micelles that co-deliver and release gemcitabine and paclitaxel in response to an acidic milieu have been proposed to selectively target stroma in the tumor core, thus achieving therapeutic effect while preserving the tumor-suppressive roles of the stroma via the peripherally located stroma [406].

Various proteolytic enzymes are involved in the active tissue remodeling observed in the PDAC TME, and their enzymatic activities can be exploited as an on-stimulus. For example, numerous MMPs have been reported to be up-regulated in PDAC, and MMPs are often considered to be key players in tumorigenesis and progression [281]. As such, MMP responsivity has been intensively studied and applied in nanomedicine [407]. PSCs have been shown to secrete various MMPs in response to interaction with pancreatic cancer cells as well as inflammatory stimuli [408,409,410]. However, a recent systematic review revealed a great discrepancy between reports regarding the expression levels of various MMPs as well as their association with clinical outcomes in PDAC [279]. Notwithstanding, most studies seem to agree that the gelatinases MMP2 and MMP9 as well as the matrilysin MMP7 are often up-regulated in PDAC among the various MMPs [279,411,412]. MMP2 responsivity has been utilized to release gemcitabine-loaded, small nanoparticles from nanofibrils in subcutaneous PDAC (Pan02 inoculated) mice [413]. MMP2-responsive liposomes to co-deliver pirfenidone and gemcitabine [414] or nanopolyplexes to deliver LY2109761 and CPI-613 (a chemotherapy agent) [415] have also been devised and tested in subcutaneous PDAC (PANC-1 + PSC co-inoculated or PANC-1 inoculated, in [414] and [415], respectively) mice. MMP-9-responsive nanovesicles that encapsulate and deliver gemcitabine have been tested on subcutaneous PDAC (PANC-1 inoculated) mice [416]. ADAMs are another class of proteolytic enzymes deregulated in PDAC; ADAMs 8, 9, and 15 have been reported to be up-regulated in PDAC [417]. Silica nanoparticles that release paclitaxel in an ADAM9-responsive manner have been devised and tested against pancreatic cancer cells in vitro [418], although functionality within the fibrotic stroma of PDAC warrants further assessment.

Redox metabolism is also deregulated in PDAC, which is driven by metabolic reprogramming and limiting levels of an important biological electron receptor, oxygen, due to intratumoral hypoxia [419]. PSCs seem to promote a more oxidized state in cancer cells and help overcome this limitation, thereby promoting cancer cell proliferation as well as conferring treatment resistance [420,421]. Glutathione biosynthesis is up-regulated in *Kras*-mutant tumors and plays an important role in alleviating redox stress [422]. A glutathione-responsive nanocomplex was devised to drive ROS generation via the Fenton reaction in tumor cells as well as macrophages. The nanocomplex, through the phenotypic reprogramming of macrophages, attenuated the myofibroblastic differentiation of fibroblasts in vitro and improved the survival of orthotopic PDAC (KPC1199 inoculated) mice [423]. The production of ROS resulting from altered redox metabolism in the PDAC TME has also been exploited in nanomedicine design. ROS-responsive polymeric micelles that co-deliver a polo-like kinase 1 inhibitor (volasertib/BI6727) together with miR-34a showed improved treatment response compared to either agent alone in orthotopic PDAC (MiaPaCa-2 inoculated) mice [424]. Micelles incorporating both ROS responsivity and acid responsivity to deliver doxorubicin have also been devised and tested in subcutaneous PDAC (PANC-1 inoculated) mice [425].

### 4.4. Strategy #8: Utilizing Nanomedicine as Nanosensitizers to Physically Manipulate the Fibrotic Stroma

We have so far focused on nanomedicine strategies that adapt to or exploit the fibrotic stroma, but nanomedicines can also be utilized to physically manipulate the fibrotic stroma directly. Given sufficient tumor-specific accumulation, the use of nanomedicines as “nanosensitizers” would allow for the locoregional physical manipulation of the PDAC TME (Figure 11C) [426,427,428].

Nanomedicines can be designed to emanate heat in response to external stimuli. For example, magnetic fields can be externally applied to induce hyperthermia (a strategy referred to as *magnetic hyperthermia*) where the nanomedicines have intratumorally accumulated [429,430]. Mild magnetic hyperthermia induced by iron oxide nanoparticles placed under alternating magnetic fields diminished the viability of PDAC cancer cells and organoids [431] and disrupted collagen fiber architecture in PDAC heterospheroids consisting of tumor cells and fibroblasts [432]. In vivo studies suggest that the co-administration of ECM-degrading enzymes such as hyaluronidase might further enhance the therapeutic effect [433]. Hyperthermia can also be induced by light (a strategy referred to as *photothermal therapy* (PTT) or *photohyperthermia*) [434,435,436,437]. An in vitro study demonstrated that the use of gold nano-rods for PTT disrupted Collagen I architecture and increased the average diffusivity of 50 nm and 120 nm nanoparticles by >10-fold [438]. Single-walled carbon nanotubes targeted with anti-IGF1R antibodies have been devised for PTT and demonstrated therapeutic efficacy against orthotopic PDAC (BxPC-3 inoculated) mice [439].

Light can also be used to activate photosensitizing molecules, which instead of emanating heat results in ROS production to kill tumor cells (a strategy referred to as *photodynamic therapy* [PDT]) [437,440]. Given systemic toxicities associated with the administration of photosensitizers such as skin phototoxicity and hyperpigmentation, nanomedicines for PDT have also been devised [435,436,441]. Nanoparticles co-delivering gemcitabine, the photosensitizer chlorin e6, and a pro-apoptotic peptide showed enhanced accumulation in tumors (compared to chlorin e6 administered as a single agent) and improved therapeutic efficacy in subcutaneous PDAC (PANC-1 inoculated) mice [442]. Liposomes delivering the photosensitizer benzoporphyrin derivative monoacid A demonstrated comparable tumor growth suppression to liposomal verteporfin (a photosensitizer) in subcutaneous PDAC (AsPC1 inoculated) mice, and the therapeutic effect further improved with the co-delivery of bevacizumab (anti-VEGF antibody) [443]. In a strategy coined *photodynamic priming* (PDP), sub-tumoricidal PDT is used not to directly kill tumor cells but to relieve fibrotic stromal barriers and improve nanomedicine delivery [444]. With PDP, the accumulation of liposomal irinotecan improved >10-fold in orthotopic PDAC (MiaPaCa-2 or AsPC-1 inoculated) mice [445]. PDP seems to exert its effect, at least in part, by affecting fibroblasts and the ECM that they produce. Using PDAC heterospheroids consisting of tumor cells and fibroblasts, a recent study demonstrated that the photodestruction of fibroblasts via verteporfin resulted in the softening of tumor tissue and increased delivery of RNA nanomedicine in vitro [446]. A photonanoimmunoconjugate consisting of lipidated benzoporphyrin derivatives conjugated to cetuximab (anti-EGFR antibody) decreased collagen density in subcutaneous PDAC (MiaPaCa-2 + CAF co-inoculated) mice [447].

In the case of both PTT and PDT/PDP, the anatomical location of the pancreas could pose technical challenges in light irradiation (for example, compared to tumors on or near the body surface). However, clinical studies have already established the safety and feasibility of therapeutic light delivery to the pancreas [448]. In contrast to PTT, PDT/PDP additionally requires molecular oxygen, which could be scarce in the hypoxic PDAC TME, to function. To circumvent limitations posed by tissue hypoxia, a nanomedicine that generates oxygen through a Fenton-like reaction from hydrogen peroxide has been devised and tested in subcutaneous PDAC (PANC-1 inoculated) mice [449].

Another external stimulus utilized is high-intensity-focused ultrasound (HIFU). In HIFU, a focused ultrasound beam is used to create thermal (hyperthermia) and/or mechanical (cavitation) effects [426,427]. HIFU applied alone induced cavitation and improved delivery and therapeutic efficacy in PDAC heterospheroids consisting of DT66066 cancer cells and normal fibroblasts [450] as well as in *KPC* mice [451]. Notably, in *KPC* mice treated with HIFU, the fibrotic stroma was disrupted, and intratumoral doxorubicin concentrations were improved by 4.5-fold [451]. HIFU can also be used in combination with microbubbles to induce cavitation and trigger drug release [452,453]. Microbubbles loaded with doxorubicin demonstrated a 12-fold improvement of intratumoral doxorubicin concentration with HIFU in subcutaneous PDAC (DSL6A inoculated) rats [454]. Interestingly, an analysis of microbubbles loaded with paclitaxel demonstrated that an optimal range of HIFU application exists and that suboptimal settings may detrimentally promote tumor growth [455]. 

## 5. Discussion: Key Unknowns and Future Directions

Having overviewed the eight main strategies to overcome fibrotic barriers to nanomedicine in the PDAC TME, we now highlight key unknowns. We furthermore propose future directions of research that are necessary to deepen our understanding of fibrotic barriers as well as to accelerate clinical translation (Figure 12). 

### 5.1. How Does the PDAC Genotype Affect the Fibrotic Phenotype?

Our understanding of the mechanisms driving fibrosis in PDAC is advancing, but there is still much left to learn. The CAF phenotype is greatly influenced by reciprocal interaction with cancer cells, whose phenotype greatly depends on genotype (mutational status) [104,312]. Indeed, treatment with the small molecule KRAS^G12D^ inhibitor MRTX1133 induced tumor regression with a concomitant increase in myCAFs and collagen deposition, which suggests that mutant KRAS plays a crucial role in shaping the composition of CAF subpopulations and the ECM [456]. Missense *TP53* mutations in PDAC cancer cells seem to promote fibrosis and shorten patient survival at least partly via promoting an immunosuppressive milieu [457]. Furthermore, *KTC* mice, which harbor pancreas-specific depletion of TGFβ signaling [458], spontaneously give rise to PDAC characterized by thicker collagen fibers, enhanced expression of fibrotic ECM proteins, as well as increased stiffness compared to *KPC* mice [101]. Because cancer cell-derived EVs can also transfer mutant proteins such as KRAS^G12D^ to non-mutant cells [459], a comparison of exosomal cargo by tumor genotype and its corresponding effects on fibrotic phenotypes might be an interesting future direction of research.

Additionally, technological advances in single-cell analyses have revealed various CAF subpopulations, but the molecular mechanisms responsible for the heterogeneity are still poorly understood [39,166]. Interestingly, gain-of-function *TP53* mutations in cancer cells seem to skew fibroblasts toward a pro-metastatic phenotype in PDAC [460]. Furthermore, mutations in breast cancer-1 (*BRCA1*) and *BRCA2* in PDAC cancer cells were recently reported to induce a myofibroblastic to immunosuppressive phenotypic change of CAFs [461]. These studies suggest that the tumor cell genotype affects the CAF phenotype, which is a concept that warrants further research. Systematically correlating the PDAC genotype with histopathology and the composition of CAF subpopulations could yield valuable information regarding the molecular mechanisms governing fibrosis progression (Figure 12A). In this regard, the expanding repertoire of GEMMs that recapitulate important genetic lesions observed in human PDAC offers numerous opportunities. Most PDAC GEMMs are based on the *Kras* oncogene [462], but approximately 8–10% of PDAC patients do not harbor *KRAS* mutations [15]. It remains to be seen whether fibrotic phenotypes and/or CAF compositions are different depending on *KRAS* mutational status. Interestingly, BxPC-3, one of the few human PDAC cell-lines that induce appreciable levels of fibrosis in murine xenografts [38], is wild type for *KRAS* and instead harbors oncogenic mutations in *BRAF* [463,464]. A PDAC GEMM harboring *Braf* mutations (*BC* mice: *Braf^CA/+^; Pdx1-CreER^T2^* and *BPC* mice: *Braf^CA/+^; LSL-Trp53^R270H/+^; Pdx1-CreER^T2^*) has also been established [465] but is relatively less employed in nanomedicine research compared to the *KPC* model. From a translational perspective, a better understanding of how the PDAC genotype affects the fibrotic phenotype will aid in patient stratification and the personalization of therapies targeting the fibrotic barriers.

### 5.2. Are CAF Subpopulations Interconvertible?

Another especially key unknown is in regard to the ambivalent role of the fibrotic stroma and the extensive heterogeneity of CAFs assumed to underlie this context-dependency [48,60,61,62,63,64,75,161,166]. A particularly important question from a translational standpoint is whether CAF subpopulations are interconvertible, not least because this is the basis of stromal reprogramming. For example, myCAFs and iCAFs seem to be interconvertible [115]. Interestingly, the TGFβ co-receptor CD105/Endoglin, abundantly expressed in PDAC stroma, has been reported to distinguish between tumor-suppressive (CD105^−^) and tumor-permissive (CD105^+^) CAFs via its effects on adaptive immunity. However, CD105^+^ and CD105^−^ CAFs seem to be non-interconvertible [466]. Perhaps this underlies the failure of CD105 neutralization to show efficacy against *KPC* mice [467]. Future research delineating distinct *lineages* (which are non-interconvertible) vs. plastic *states* (which are interconvertible) of CAFs is thus warranted (Figure 12B) [75,166]. 

What determines interconvertibility (or lack thereof) between CAF subpopulations? While this is a question open to future research, epigenetic cell memories might be at play [468]. Different cell-of-origins may impart distinct epigenetic and thus transcriptional states to CAFs. CAFs also dynamically respond and adapt to their surrounding microenvironment during their life cycle [469]. For example, prolonged exposure to a stiff microenvironment (such as found in the PDAC TME) causes irreversible phenotypic changes to fibroblasts and mesenchymal stem cells, which is a phenomenon known as *mechanical memory* [470]. Mechanical memory induces epigenetic remodeling that causes persistent changes in the transcriptome, such as the expression of miR-21 that preserves mechanical memory [471,472,473]. In addition to the mechanical microenvironment, contact with tumor cells can also induce genomic methylations in genes such as *SOCS1* in PDAC CAFs [474]. Moreover, lactate derived from PDAC tumor cells drives widespread epigenetic reprogramming of CAFs through the activation of TET demethylases via stimulating increased production of α-ketoglutarate (a cofactor for TET enzymatic activity) [56]. These findings suggest that CAFs preserve “memories” of their microenvironment via epigenetic mechanisms [47,475]. Indeed, there is increasing recognition of the importance of epigenetics in fibrotic processes [476,477,478]. It would be interesting to analyze the relationship between the interconvertibility of CAF subpopulations and their epigenetic states. With an expanding arsenal of chemical modulators of epigenetic processes [479,480] and an increased understanding of the transcriptomic landscape of different CAF subpopulations [163,164,165,166,167,168], non-interconvertibility might be therapeutically overcome in the future [481,482].

### 5.3. What Distinguishes Good from Bad Stroma?

Given the association of bulk stromal expression signatures with patient prognosis [372], one wonders how different CAF subpopulations existing in various ratios within an individual tumor dictate the behavior of the stroma as a whole (Figure 12C). Clinically useful indices that distinguish between good and bad stroma have recently begun to emerge [75]. Knowledge of the underlying mechanisms would be useful in defining strategic objectives when therapeutically targeting the fibrotic stroma. This is especially pertinent considering that it is currently not entirely clear what objectives a successful stromal reprogramming strategy needs to achieve. Moreover, further identification of discriminative biomarkers of good vs. bad stroma could aid in patient stratification, which is key in advancing the clinical application/translation of nanomedicine [62,483]. 

Notably, the TME is highly heterogeneous even within a single patient: a cursory look at fibroblast morphology reveals great variation [75]. Interestingly, it has recently been reported that the PDAC TME consists of sub-compartments (sub-TMEs) characterized by distinct fibroblast morphologies (*deserted* regions with spindle-shaped CAFs with abundant ECM and low cellularity, *reactive* regions with plump CAFs with enlarged nuclei, and regions with *intermediate* characteristics) that possess distinct immune phenotypes as well as response to chemotherapy [484]. Because the different sub-TMEs possess distinct expression profiles, bulk RNA sequencing data of tumor tissue could be utilized to predict sub-TME status and survival differences [484]. While the implications of sub-TMEs for nanomedicine are currently unclear, the abundance of ECM in deserted vs. reactive regions presumably will affect the efficiency of nanomedicine penetration. A better understanding of how the different CAF subpopulations and sub-TMEs arise, interact, and evolve throughout tumorigenesis and treatment will advance our understanding of fibrosis in PDAC and inform novel therapeutic strategies targeting the fibrotic stroma [161,173].

### 5.4. How Does Stromal Tissue Architecture Affect Nanomedicine Delivery?

A related key unknown is how stromal composition and tissue architecture govern tumor mechanics and vice versa. We are only beginning to understand how signaling at a cellular scale affects tissue architecture [485,486]. We know even less about how stromal tissue architecture, in turn, affects nanomedicine delivery and therapeutic efficacy (Figure 12D). How the tissue architecture and mechanics of the TME dynamically evolve over time needs to be studied. Indeed, fibroblast and ECM composition both dynamically change in response to chemotherapy, which likely affects tumor mechanics and seems to entail treatment resistance [173,484,487]. To understand the progression of fibrosis, some have suggested the need for an atlas of fibrotic tissue: a spatially and temporally co-registered dataset of ECM mechanical properties, composition, and organization, and various relevant parameters of cell biology acquired at both micrometer and millimeter resolution [488].

The failure of stromal ablation should serve as a cautionary tale: any therapeutic intervention that modulates the stroma will alter tumor mechanics which may have therapeutic consequences [101]. While altering stromal tissue architecture against a common genetic background is extremely difficult, if not impossible, in vivo; a bottom–up approach to modeling the PDAC TME in vitro is increasingly enabling the generation of PDAC tissues with different stromal architectures [35,36,238]. Together with an in silico, mathematical modeling of nanomedicine delivery [489,490], studies using advanced in vitro models of PDAC will likely shed light on the relationship between stromal tissue architecture and the efficacy of nanomedicine delivery.

### 5.5. How Do Nanomedicines Penetrate the PDAC Stroma?

Another unknown is the route of nanoparticle penetration of PDAC stroma (Figure 12E). While paracellular passage is often assumed as the main route of nanomedicine passage through PDAC stroma, experimental evidence is scarce. CAFs have been shown to interact with and internalize nanoparticles [491], which affects the efficacy of nanomedicine penetration [33]. The engagement of transcellular/transcytosis pathways is mainly being studied to improve the extravasation of nanomedicine [492,493,494] but might also improve delivery through the fibrotic stroma. 

The relative contributions of the various cellular and ECM components comprising the fibrotic stroma remain difficult to parse, although advanced in vitro models are increasingly enabling the experimental tuning of various variables that would otherwise be difficult to directly manipulate in vivo. For example, because ECM components are central to normal development, knockout of ECM protein-encoding genes is often embryonic lethal [495]. A lack of CAF-specific *Cre*-driver lines as well as extensive heterogeneity in fibroblast transcriptomes, greatly driven by ECM/matrisome genes, can also complicate conditional knockout strategies [48,496]. On the other hand, in vitro models would allow the spatially and temporally controlled manipulation and analysis of individual ECM genes. The expanding ability to model and tune physicochemical parameters of the tumor tissue (such as stiffness) in vitro, for example through the use of advanced hydrogels [497,498], also holds great promise in advancing our understanding of the TME.

### 5.6. What Are the Mechanisms Governing Nanoparticle-Bio Interactions?

From a nanotechnology perspective, a key unknown is the nanoparticle–bio interface of nanomedicines: from the moment of administration until the release of the payload against the cancer cells, nanomedicines complexly interact with the body (Figure 12F) [22,321,325,499,500,501]. This is an especially important issue in clinical translation, since the nanoparticle–bio interface determines the safety and efficacy of nanomedicines. Prime examples are corona formation on the surface of nanomedicines affecting its functionality [382,385] and the various ancillary effects of nanoparticles [325]. Conventional nanomedicine design criteria based on an over-simplistic understanding of the nanoparticle–bio interface have been argued to underlie the low success rate of clinical translation [502]. A fuller understanding of the nanoparticle–bio interface will allow the principled, rational design of nanomedicines which is necessary to accelerate clinical translation [324]. The development of high-throughput screening systems, combined with machine learning approaches, to systematically correlate nanoparticle properties with various biological readouts would be one way forward [503,504,505]. An alternative, but complementary, approach would be to study endogenous systems. Most, if not all, cell types secrete and employ EVs to communicate [506,507]. EVs including exosomes are increasingly implicated in fibrogenesis [508,509]. An interesting aspect of EVs is their tropism (cell- and/or tissue-specificity of uptake) [510]. The tropism of EVs at least partly seems to be dictated by the specific type of integrin heterodimer present on their surfaces [511]. An improved understanding of EV tropism could perhaps inspire novel active targeting methods with enhanced tissue specificity. EVs can also be directly exploited as therapeutics and may be advantageous for clinical translation in terms of safety [355,512].

Another important point is that our current understanding of the nanoparticle–bio interface is based mostly on animal studies [321]. From a translational perspective, the assessment especially of long-term safety and efficacy in humans is key for cancer nanomedicine to be a sustainable strategy in the clinic [324]. The emergence of immune responses against nanomedicines upon repeated exposures and the associated diminution of therapeutic efficacy is a case in point [513,514]. Increased understanding of these long-term responses will not only ensure patient safety but can also motivate the development of new materials [515,516].

### 5.7. How Can Therapeutic Strategies Targeting Fibrotic Stroma Be Utilized in Combination Therapies?

The fibrotic stroma is not just a physical barrier to nanomedicine delivery but itself an important determinant of clinical outcome [372,484]. Interestingly, the same TME features that result in poor efficiency of nanomedicine penetration are also immunosuppressive [517]. Indeed, fibrosis driven by TGFβ signaling promotes immune evasion in PDAC [114]. A pan-cancer transcriptomic analysis revealed that TGFβ-associated ECM gene signatures correlate with immune evasion [518]. Inhibition of collagen cross-linking softens tumor tissue and improves T cell migration as well as anti-PD-1 treatment in *KPC* mice [243]. Thus, therapeutic strategies targeting the fibrotic barriers to nanomedicine penetration will likely synergize with immunotherapeutic strategies [517]. This is an exciting avenue of research given the recent, widespread clinical deployment of mRNA vaccines based on decades of basic research but which progressed with the COVID epidemic as a final impetus [519,520]. Indeed, mRNA cancer vaccines against PDAC are in clinical development [15,521]. Whether therapeutic strategies targeting fibrotic barriers potentiate vaccine efficacy against PDAC is an interesting question. More generally, an improved understanding of the pathobiology of fibrosis in PDAC will enable us to move forward from an ad hoc to a rational approach to designing combination therapies (Figure 12G) [483].

### 5.8. Which Experimental Model Should Be Used to Study Fibrotic Barriers in PDAC?

Lastly, we would like to emphasize that the choice of the experimental model of PDAC used, whether in vitro or in vivo, is especially crucial when studying fibrotic barriers (Figure 12H) [32,501]. In animals, this is because different models recapitulate fibrotic lesions to varying extents, which may have direct implications for the potential, clinical translation of research findings [502]. Numerous commonly used human PDAC cell lines fail to demonstrate appreciable levels of fibrosis when xenografted into mice. A notable exception is the BxPC-3 cell-line, especially when inoculated in the presence of FGF2 [38]. Co-inoculation with fibroblasts/PSCs is thus usually necessary to model fibrosis in cell-line derived PDAC xenografts in mice [522]. On the other hand, GEMMs tend to recapitulate fibrotic histopathology better but are time-consuming and relatively costly [386,462,523,524]. Whatever the model used, histological analyses should be routinely performed to assess fibrosis in animal models of PDAC to correlate histopathology with nanomedicine delivery and efficacy. 

Regarding in vitro models of PDAC, 3D cell culture models are increasingly utilized to more faithfully recapitulate pathogenetic mechanisms as well as to predict therapeutic outcomes [523,524,525,526,527]. Advances in organoid technology are particularly remarkable [523,528]. That PDAC patient-derived organoids recapitulated clinical responses to treatment in individual patients is a testament to their clinical relevance [529]. The incorporation of stromal elements (e.g., vasculature and fibrosis) remains a challenge in many 3D cell culture models but will open up various opportunities to study the TME [524,526,528]. The characterization of 3D cell culture models in terms of tumor mechanics and treatment response as compared to PDAC in vivo is lagging and warrants thorough analyses in the future.

## 6. Conclusions

Overcoming fibrotic barriers is necessary for nanomedicine to be effective against PDAC. As detailed above, numerous therapeutic opportunities exist. The fibrotic stroma may be therapeutically modified through stromal ablation (strategy #1), stromal reprogramming (strategy #2), targeting fibroblast metabolism (strategy #3), and/or targeting ECM abnormalities (strategy #4). In addition, optimization/tuning of nanomedicines will also likely be required. This includes the optimization of various physicochemical properties (strategy #5), active targeting strategies (strategy #6), installation of microenvironmental responsivity (strategy #7), and/or the use of nanomedicines as nanosensitizers (strategy #8). So far, targeting fibrosis in PDAC has proven more difficult than initially appreciated, and the field has experienced numerous disappointments. We are always learning something new: our improved understanding of fibroblast and stromal biology will continue to inform novel therapeutic strategies to ameliorate fibrotic barriers to nanomedicine penetration in the PDAC TME. It is worth mentioning that just 10 years ago, CAF activation was deemed an irreversible process [46], which shows just how far our understanding has progressed. All the more since the effects of fibrosis on weakening treatment response are not limited to nanomedicine: strategies that successfully target fibrosis hold great promise in improving the prognosis of patients with PDAC.

## Figures and Tables

**Figure 1 cancers-15-00724-f001:**
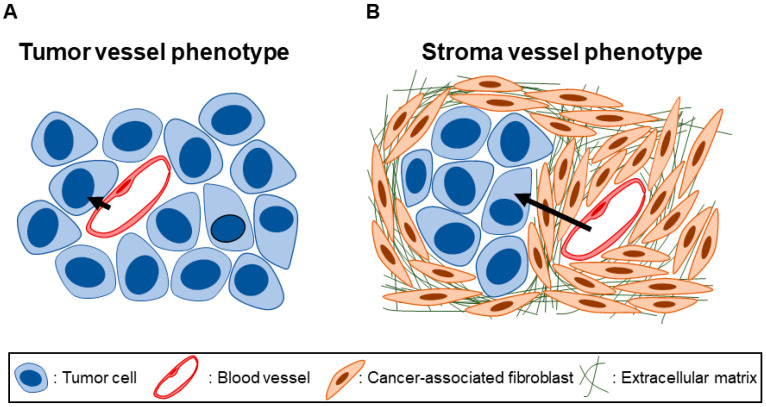
Tumor vessel phenotype vs. stroma vessel phenotype. (**A**) In tumor tissues with a tumor vessel phenotype, blood vessels are situated close to tumor cells with not much stromal tissue in between. (**B**) In tumor tissues with a stroma vessel phenotype, blood vessels are distanced from the tumor cells by stromal tissue. In the case of pancreatic ductal adenocarcinoma (PDAC), the stroma consists of dense, fibrotic tissue comprising cancer-associated fibroblasts (CAFs) and deposited extracellular matrix (ECM) proteins. Note the longer path (black arrows) an intravenous nanomedicine must travel to reach tumor cells in tissues with a stroma vessel, as opposed to a tumor vessel, phenotype.

**Figure 2 cancers-15-00724-f002:**
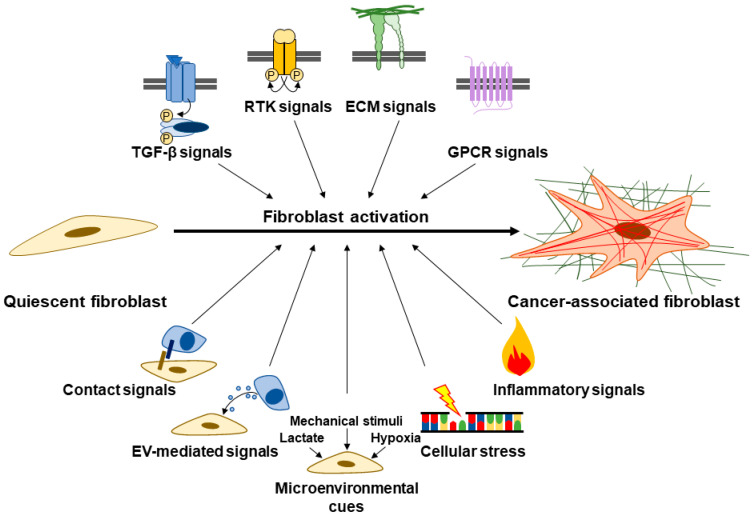
Signals involved in fibroblast activation. Quiescent fibroblasts become activated into CAFs, which are traditionally characterized by α-smooth muscle actin (αSMA) positivity and abundant ECM secretion/deposition. Fibroblast activation is driven via various signals such as transforming growth factor-β (TGFβ), receptor tyrosine kinase (RTK) signals, ECM signals, G-protein coupled receptor (GPCR) signals, contact signals, extracellular vesicle (EV)-mediated signals, microenvironmental cues, cellular stress, and inflammatory signals.

**Figure 3 cancers-15-00724-f003:**
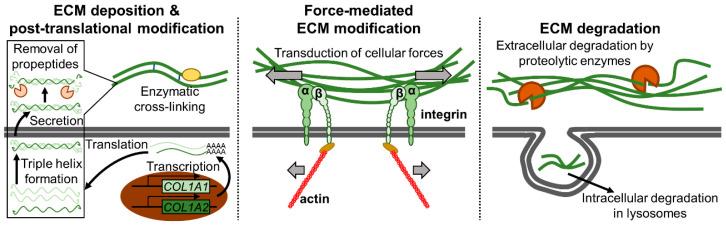
The ECM remodeling process. The three stages of ECM remodeling: (1) ECM deposition and post-translational modification (**left**), (2) force-mediated ECM modification (**middle**), and (3) ECM degradation (**right**) are illustrated with collagen I as an example. In the ECM deposition and post-translational modification step, individual collagen chains are first transcribed/translated. The collagen chains then form a triple helix, after which they are secreted, and the propeptides necessary for triple helix formation are removed. Secreted collagen fibers undergo various modifications such as enzymatic cross-linking. In the force-mediated ECM modification step, cellular forces are transduced via ECM receptors such as integrins to deform and align collagen fibers. Finally, in the ECM degradation step, collagen fibers are degraded extracellularly via various proteolytic enzymes or intracellularly in lysosomes following internalization.

**Figure 4 cancers-15-00724-f004:**
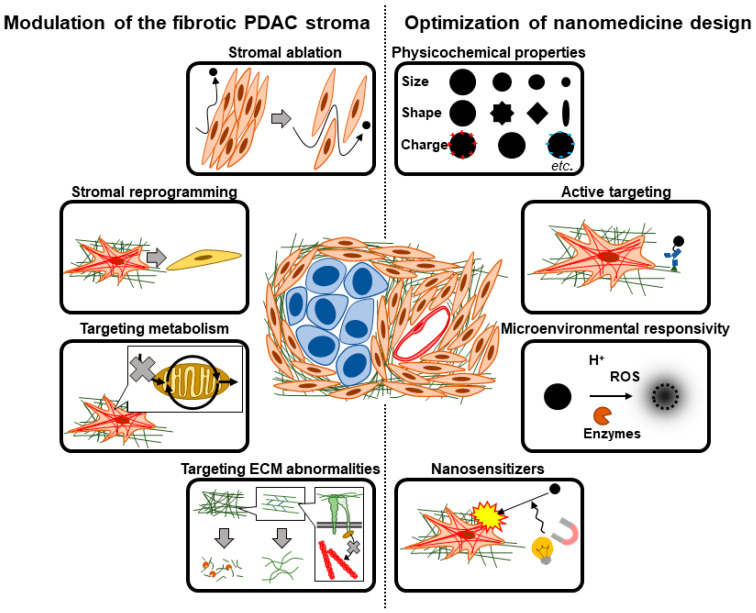
Therapeutic strategies to overcome fibrotic barriers to nanomedicine. To overcome the fibrotic barriers to nanomedicine in the PDAC tumor microenvironment (TME), the fibrotic stroma could be modulated (**left**) and/or the nanomedicine design could be optimized (**right**). Strategies to modulate the fibrotic PDAC stroma include stromal ablation, stromal reprogramming, targeting aberrant CAF metabolism, and targeting ECM abnormalities. Strategies to optimize nanomedicine design include tuning of physicochemical properties, installation of moieties for active targeting and/or microenvironmental responsivity, as well as the use of nanomedicine as nanosensitizers for locoregional stromal modulation.

**Figure 5 cancers-15-00724-f005:**
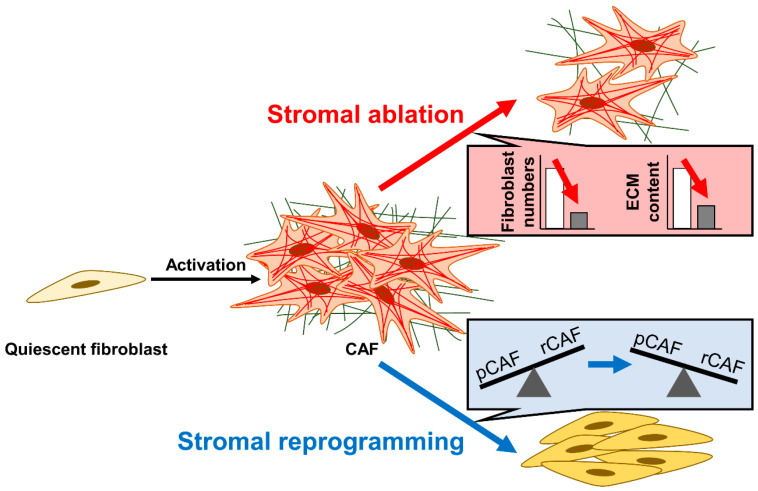
Stromal ablation vs. stromal reprogramming. Stromal ablation (red arrow) aims to reduce stromal content by decreasing fibroblast numbers and ECM content. In contrast, stromal reprogramming (blue arrow) aims to shift the balance from CAFs with tumor-promoting functions (pCAFs) to CAFs with tumor-restraining functions (rCAFs). The insets schematically depict the objectives of each strategy.

**Figure 6 cancers-15-00724-f006:**
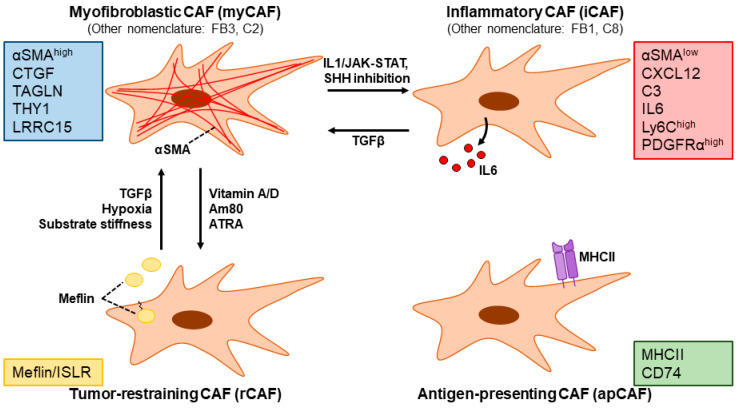
CAF subpopulations in PDAC and representative molecular markers. Myofibroblastic CAFs (myCAFs, also referred to as the FB3 or C2 subpopulation) are characterized by the high expression of αSMA. Inflammatory CAFs (iCAFs, also referred to as FB1 or C8 subpopulation) are characterized by the low expression of αSMA as well as the expression of inflammatory cytokines such as interleukin (IL) 6. rCAFs are characterized by the expression of Meflin/ISLR. Antigen-presenting CAFs (apCAFs) express MHC class II and CD74. TGFβ signaling as well as hypoxia and substrate stiffness has been shown to induce a myCAF phenotype. IL1 signaling through JAK-STAT as well as sonic hedgehog (SHH) signaling inhibition induces a myCAF-to-iCAF conversion. Vitamin A/D, the synthetic retinoid Am80, and all-*trans* retinoic acid (ATRA) promote the induction of the rCAF subpopulation. Note that the figure illustrates some of the more well-characterized CAF subpopulations to date and is not meant to be an exhaustive depiction.

**Figure 7 cancers-15-00724-f007:**
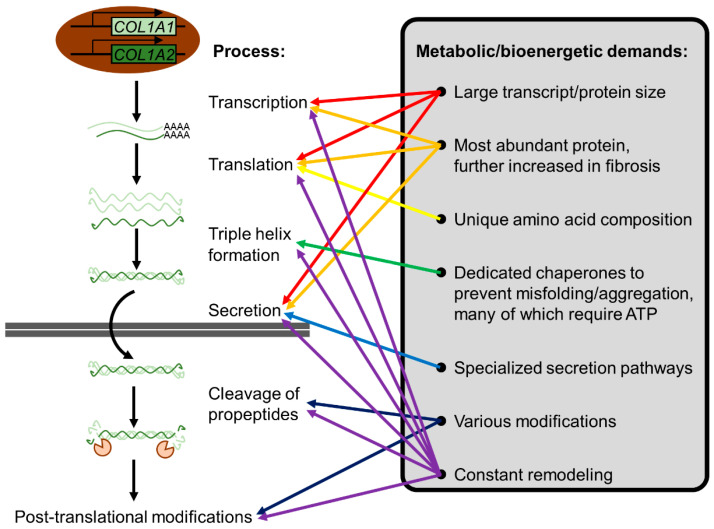
The metabolic/bioenergetic demands of ECM production. Using collagen I as an example, the metabolic/bioenergetic demands inherent to ECM production are illustrated. Its large size, abundance, unique amino acid composition, and tertiary structure pose a unique challenge for cells. Not only is transcription/translation metabolically costly, but chaperones to prevent misfolding/aggregation are also required. Moreover, collagens are secreted via specialized pathways due to their large size [204]. Various post-translation modifications and constant remodeling further increase metabolic/bioenergetic costs.

**Figure 8 cancers-15-00724-f008:**
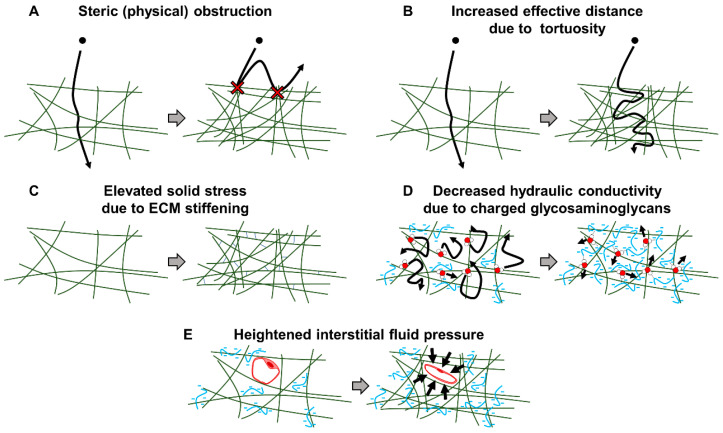
The ECM barrier to nanomedicine delivery. The ECM poses a barrier to nanomedicine delivery via various mechanisms. (**A**) The ECM can sterically obstruct nanomedicine passage. (**B**) The increased deposition of ECM as well as its compression from neighboring cells increases the tortuosity of the ECM network. This results in an increased effective distance a nanomedicine must travel to locate a tumor target. (**C**) Cross-linking causes stiffening of the ECM network which elevates solid stress. (**D**) Negatively charged glycosaminoglycans such as hyaluronan trap water and reduce hydraulic conductivity. (**E**) Increased ECM deposition and trapped water result in elevated interstitial fluid pressure which causes vascular compression/collapse.

**Figure 9 cancers-15-00724-f009:**
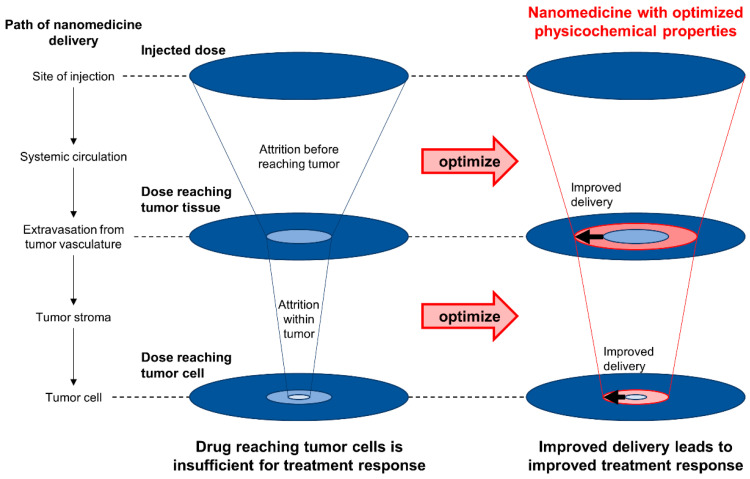
Optimizing physicochemical properties of nanomedicine to improve delivery and treatment response. A significant amount of the injected nanomedicine is lost before it reaches the tumor cells to elicit a therapeutic effect. By optimizing the physicochemical properties of nanomedicine such as particle size, particle geometry, surface charge, surface chemistry, and elasticity, the efficiency of nanomedicine delivery can be improved. Optimization must not only address attrition due to fibrotic barriers but also attrition even before the nanomedicine reaches the tumor tissue.

**Figure 10 cancers-15-00724-f010:**
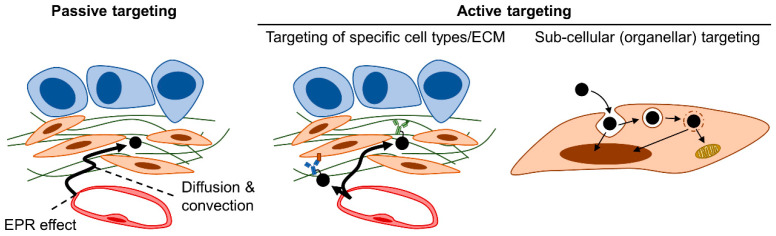
Passive vs. active targeting. Passive targeting of nanomedicine relies on extravasation via the enhanced permeability and retention (EPR) effect followed by diffusion and convection. In active targeting, moieties (antibody is depicted as an example in the figure) that recognize specific features of the tumor tissue (such as cell surface markers or ECM components) are utilized to enhance accumulation. Active targeting also encompasses sub-cellular (organellar) targeting, which is the targeted delivery of the payload to a specific organelle/location after internalization by the target cell.

**Figure 11 cancers-15-00724-f011:**
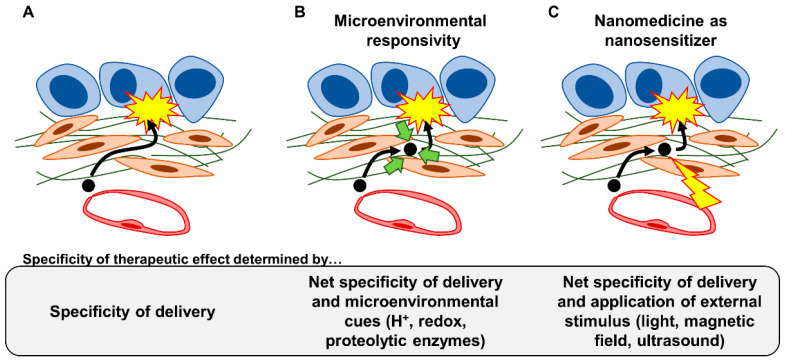
Increased specificity of the therapeutic effect via microenvironment-responsive nanomedicines and nanomedicines as nanosensitizers. While the specificity of the therapeutic effect of conventional nanomedicines is determined by the specificity of delivery (**A**), the installation of responsivity to microenvironmental cues (light green arrows) allows an additional layer of control over specificity (**B**). Nanomedicines could also be designed to respond to an external stimulus (yellow thunderbolt), in which they could be considered “nanosensitizers” (**C**).

**Figure 12 cancers-15-00724-f012:**
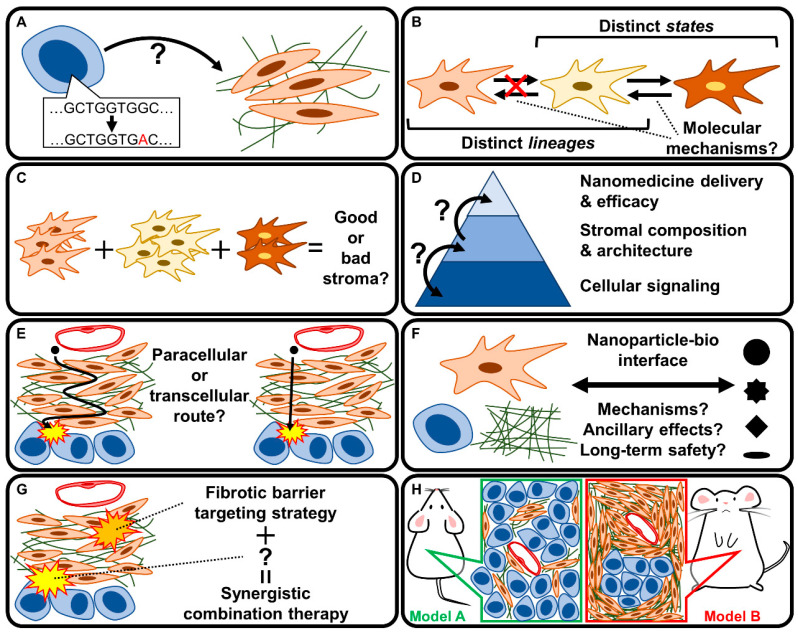
Key unknowns and future directions. (**A**) How does the genotype of the PDAC cancer cell affect fibrosis? (**B**) Are CAF subpopulations interconvertible, and what determines interconvertibility? (**C**) What distinguishes good from bad stroma, and how does a particular mix of CAF subpopulations determine stromal behavior? (**D**) How do stromal composition and architecture affect nanomedicine delivery and efficacy? How do signaling at the cellular level and stromal composition/architecture interrelate? (**E**) How do nanomedicines penetrate the tumor stroma? What is their route of passage? (**F**) What are the mechanisms governing the nanoparticle–bio interface, and how do these mechanisms affect the ancillary effects of nanomedicine? Are the nanomedicines safe in the long term? (**G**) What can be combined with strategies targeting the fibrotic barrier to achieve synergy? (**H**) Which model should be used to study fibrotic barriers?

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
