# Peer review of "Therapeutic Strategies to Overcome Fibrotic Barriers to Nanomedicine in the Pancreatic Tumor Microenvironment"

_cancers, 2023, doi:10.3390/cancers15030724_

Round 1

Reviewer 1 Report

The authors summarize the strategies to overcome fibrotic barriers for nanomedicines in the pancreatic tumor environment. This manuscript fell within the scope of Cancers, and was well organized. Over 500 papers were referred to. I recommended a Minor Revision before final acceptance.

Detailed comments:

Q1-Nine keywords might be too many, and 4~6 keywords were most common.

Q2-As this manuscript was summarizing the strategies associated with nanomedicines, I suggested to make a brief summary about the potential nanomedicines for pancreatic tumor treatment in the Introduction Section.

Q3-Please unify the uppercase/lowercase of the first words in all figures.

Q4-For Section 5, could the authors add some discussion about the clinical translation? Cancers is a flagship journal concerning the field of clinical oncology.

Q5-The current Conclusion Section was more like a ‘concluding remarks’. It was suggested to firstly summarize the strategies in the text in the Conclusion Section.

Q6-A minor advice: Many foundations were acknowledged in the Funding Section. As a REVIEW, was it necessary to consume so much financial funding?

Reviewer 2 Report

This is a comprehensive and considered review of the issues concerning the successful application of nano medicines to PDAC. It is well written and well organised, following a natural progression from a description of the PDAC stroma, through its impact on therapeutics and targeting, to the implications for possible strategies for overcoming the barriers it presents to nanoparticle therapeutics. Moreover, it considers the limits to our current knowledge, what needs to be known for effective application of nano medicines to PDAC and how this would be best modelled / achieved.

The review is not novel in that other reviews have considered how to optimise therapeutics, including nanoparticle therapeutics for targeting PDAC. However, this review stands out in the comprehensive way the sub-topics are integrated and lead to future directions including their discussion.

I did consider whether the second section, essentially a background description of the molecular cell biology of the PDAC stroma was necessary in that it borders on general knowledge reviewed elsewhere. Indeed, the reader is referred elsewhere to reviews for other relevant specific topics for example the stroma metabolism and aerobic respiration. However, I felt the value here is greater for having all of the topics together in the integrated format.

Only one issue requiring attention, the opening sentence in the Simple Summary section is quite general and not particularly accurate. Of the "various' cancers, pancreatic cancer, though very serious, doesn't have the worst survival rate. Glioblastoma for example, is worse. An appropriate adjustment to the text should be made. Of course, there are many ways that severity can be expressed, total deaths, disease free progression etc. Something that conveys the severity but without being misleading is required.
